# RDSC: Range-Based Device Spatial Clustering for IoT Networks

**DOI:** 10.3390/s24175851

**Published:** 2024-09-09

**Authors:** Fouad Achkouty, Laurent Gallon, Richard Chbeir

**Affiliations:** 1OpenCEMS, LIUPPA, E2S UPPA, University Pau & Pays Adour, 64600 Anglet, France; fouad.achkouty@univ-pau.fr; 2OpenCEMS, LIUPPA, E2S UPPA, University Pau & Pays Adour, 40000 Mont de Marsan, France; laurent.gallon@univ-pau.fr

**Keywords:** clustering, spatial data, IoT, coverage, capacity-aware

## Abstract

The growth of the Internet of Things (IoT) has become a crucial area of modern research. While the increasing number of IoT devices has driven significant advancements, it has also introduced several challenges, such as data storage, data privacy, communication protocols, complex network topologies, and IoT device management. In essence, the management of IoT devices is becoming more and more challenging, especially with the limited capacity and power of the IoT devices. The devices, having limited capacities, cannot store the information of the entire environment at once. In addition, device power consumption can affect network performance and stability. The devices’ sensing areas with device grouping and management can simplify further networking tasks and improve response quality with data aggregation and correction techniques. In fact, most research papers are looking forward to expanding network lifetimes by relying on devices with high power capabilities. This paper proposes a device spatial clustering technique that covers crucial challenges in IoT. Our approach groups the dispersed devices to create clusters of connected devices while considering their coverage, their storage capacities, and their power. A new clustering protocol alongside a new clustering algorithm is introduced, resolving the aforementioned challenges. Moreover, a technique for non-sensed area extraction is presented. The efficiency of the proposed approach has been evaluated with extensive experiments that gave notable results. Our technique was also compared with other clustering algorithms, showing the different results of these algorithms.

## 1. Introduction

The Internet of Things (IoT) is a network of objects that are connected, providing data collection, data analysis, and decision making. The number of IoT devices is increasing each year. IoT is becoming increasingly widespread in many domains and contexts like health care, smart homes, smart cities, agriculture, energy optimization, supply chains, and environmental monitoring [1].

The proliferation of connected IoT devices has led to numerous challenges across different domains:Security is one of the major concerns, as the large number of devices connected to the network makes it more vulnerable to attacks by increasing the potential entry points for cybercriminals.Data management also represents a significant challenge. The massive amounts of data collected by IoT devices must be stored in data centers for processing and analysis, which can be costly in terms of storage, financial resources, and human administration. Moreover, these sensitive data must be anonymized and, in some cases, encrypted to preserve user privacy and confidentiality.Another key challenge is the heterogeneity of IoT devices. These devices are produced by different manufacturers, with varying sensing capabilities, power consumption, and storage capacities, which complicates their management.Finally, the management of IoT networks also brings its share of complexities, particularly when devices use distinct communication protocols, such as Wi-Fi and Bluetooth, making their interconnectivity more difficult.

Diverse solutions for IoT management were developed. For instance, edge computing [1] allows the processing of data closer to its source, reducing network load and latency. Another solution that has been developed is automated device management, particularly during the deployment process. This involves implementing techniques for auto-configuration and automated maintenance of IoT devices. Scalability in network architectures has been addressed by designing robust network architectures that are scalable and can respond to network dynamics and failures. Additionally, energy-efficient protocols that track device energy and battery depletion and charging have emerged.

This paper highlights several key challenges, including the constrained capacities of the devices, the limited sensing area of the devices, and the power optimization of the according IoT devices. Here, we propose a novel solution for managing the diverse landscape of IoT devices. Our approach groups IoT devices with varying capabilities while considering power consumption, and organizes them into a hierarchical clustering architecture. This hierarchical clustering design enables several benefits for IoT device management, including data aggregation, load balancing, and increased network availability. By intelligently grouping and hierarchically managing heterogeneous IoT devices, our solution addresses key challenges in large-scale IoT deployments, enabling more efficient and reliable device management. A comparison was made between RDSC and other clustering techniques (k-means and DBSCAN). The comparison demonstrated that these clustering algorithms resulted in overlapping big clusters without considering the device storage capacities and the residual energy. For this reason, we propose the RDSC approach, which clusters devices based on their coverage range, eliminates overlaps, and takes into account both device storage capacities and residual power while clustering, emphasizing the novelty of our approach.

The rest of this paper is organized as follows: In the next section, we present a motivating scenario highlighting IoT characteristics, emphasizing the addressed challenges. In Section 3, we present related work featuring a background about clustering, specific device clustering use cases and finalized with a comparison table. After that, we will present some preliminaries and assumptions (Section 4) used further. In Section 5, we detail our approach along with our clustering protocol. Section 6 explains our clustering algorithm before we detail in Section 7 the uncovered zone generation. Section 8 shows the set of experiments that concerns the clustering algorithm execution before we conclude this study in Section 9.

## 2. Motivating Scenario

To provide context and motivation for our proposal, we will consider the use case of devices deployed in the Chiberta forest of Anglet, France.

### 2.1. Chiberta Forest Setup

In the Chiberta forest, an environmental enterprise deployed devices equipped with infrared-based temperature sensors to prevent fires in the forest and conduct analysis. The enterprise deploys devices randomly throughout the forest, as some areas are easily accessible (plain fields), while others are not (hills and mountains). This random distribution results in varying device densities in the forest. Easily accessible areas have a high device density, while isolated areas are almost empty. Infrared sensors are non-contact sensors that can measure temperature within a specific coverage range. The coverage range can vary depending on the sensor type, the detector sensitivity, and the intensity of the emitted radiation. Typically, infrared sensors can measure distances ranging from a few centimeters to several meters.

### 2.2. Device Heterogeneity

The deployed devices are heterogeneous (i.e., sensing attributes of different types) and have different properties, as shown in Figure 1a. First, they have different coverage ranges since they are provided by different industrial companies. Second, devices have varying and limited storage capacities. Last but not least, devices have limited power consumption, which impacts the network performance and its overall lifetime.

### 2.3. Challenges

In fact, extending the lifespan of an IoT network is essential for providing long-term reliability and avoiding network shutdowns. A lot of strategies help in achieving network lifetime expansion:-Energy-efficient hardware: using devices with low energy consumption while enabling sleep/wake methodologies will expand the network’s life expectancy. In addition, using energy harvesting techniques with solar panels and other energy sources will keep devices alive for a longer duration.-Data aggregation: aggregating the data (crowd wisdom) and forwarding them at once will reduce the number of devices involved and extend the network’s lifetime. Furthermore, data aggregation offers functionalities that are unavailable when each device operates independently, such as handling missing data, detecting anomalies, and ensuring data quality.-Communication Protocols: efficient transmission protocol (e.g., MQTT) will reduce the number of devices involved during the packet transmission, hence enhancing the network lifetime.-Load balancing: load balancing involves distributing network load across multiple nodes. By spreading tasks among different nodes, heavy workloads can be divided, reducing resource consumption and, consequently, energy consumption.

Additionally, the network topology has a significant effect on the performance of the network. Leaving each device working on its own will increase the network load and complexity since each node will have to return its data to the sender (user, base station). Furthermore, in connected environments, devices can go down easily due to weather conditions or a false configuration, making network recovery challenging in that case. To bypass this difficulty and increase network efficiency, a common practice has been adopted in the Chiberta forest, which consists of grouping the devices together while choosing a responsible device for each group (as shown in Figure 1b). In essence, grouping devices will lead to data load balancing between the devices (blue devices, example), network stability in case of a device failure (brown devices, for example), and network scalability in case of a newly added device (red devices, for example). The coverage area is a crucial factor to consider when grouping the devices because it guarantees that all of the devices’ sensing areas cover the target region without any noticeable overlaps or gaps, maximizing the network’s monitoring and data collection efficiency and effectiveness. We note that we define the coverage area as the sensor’s sensing area and not its communication range. For the sake of simplification and ease of understanding, devices are assumed to be sensing the same observation (i.e., temperature).

Thus, to manage these issues, various challenges need to be addressed:Challenge 1: How to cluster devices while taking into consideration their limited storage capacity?Challenge 2: How to manipulate device coverage range while clustering? How to manipulate coverage range gaps while clustering?Challenge 3: How to take into account device power while clustering to optimize network lifecycle?Challenge 4: How can device connectivity be considered while clustering to optimize packet forwarding between clusters?

It is important to note that, in large connected environments, network behaviors are unpredictable due to their complexity. Choosing query destinations in an IoT network can streamline the network, simplifying many networking tasks and reducing overloads and data flows. The physical network architecture can also facilitate query routing directly to the destination, such as mesh networks and ad hoc networks. In our use case, each query is directly routed from the sender (external user) to the appropriate cluster head (through either a single-hop or multi-hop path), as the query destinations are predetermined at the time of issuing the query (collecting information from a specific area such as cities or forests). Therefore, network connectivity challenges (challenge 4) will be addressed in a future study.

## 3. Related Works

In this section, we will detail some studies related to clustering and more specifically device clustering.

### 3.1. Clustering Background

In IoT, device clustering has many benefits and can make major differences in network performances. A clustering survey for k-means and other clustering algorithms was conducted in [2]. In this survey, the authors described many clustering methodologies and technologies:Centroid-based: Objects are assigned to the nearest cluster head based on the distance between the current point and other cluster heads (CHs). Some examples of centroid-based algorithms are the k-means and k-medoids. These algorithms are used in many use cases related to energy management (electric vehicles [3] and smart grids [4]), network security (false data injection [5]), and the identification of unstable cluster heads [6]).Distribution-based: These clustering algorithms rely on the probabilistic distribution of the objects. The clustering model calculates the probability of an object being assigned to a specific cluster. Gaussian mixture model (GMM) clustering is an example of a distribution-based algorithm. This technique can be used to model electricity consumption patterns [7,8] and perform system reliability analysis [9]. Another popular distribution-based technique is Bayesian clustering. Bayesian clustering can be used for model parameter estimation [10,11] and energy consumption pattern detection [12].Density-based: The goal of such algorithms is to group objects with high density. These algorithms are suitable for complex data with different shapes and structures. DBSCAN is a popular example of density-based algorithms. This type of methodology is used in anomaly detection and in the discovery of power consumption patterns [13].

There are also other clustering techniques (grid-based, graph-based, shapelet-based) that also partition data and aim to find semantic relations between the data tuples.

At the end of the clustering process, depending on the technique, each cluster can have a center point named centroid. The centroid can be any point in the data space, or it can be an actual node of the cluster. In networking, having a node that manages others can improve network performance by performing local operations (data aggregation and correction). In other words, having an actual node acting as a centroid is crucial in networking. This “manager” node is named cluster head. The dependency between CH and cluster members (CM) exhibits a hierarchical relationship between these nodes, leading to the establishment of a hierarchical architecture.

A type of clustering capable of acquiring a hierarchical architecture is the hierarchical clustering. The result of this clustering type will be a hierarchical tree of devices, each group of devices having a cluster head responsible of the entire group. In hierarchical clustering, there are two main types of algorithms: divisive and agglomerative. In divisive hierarchical clustering, all the nodes are grouped into a single cluster; then they are divided to form smaller clusters. It is used to break down large-scale data [14] and transform them into subsets of data, simplifying load management [15]. In agglomerative hierarchical clustering, a bottom–up methodology is employed, where each member starts as an independent cluster, and then cluster pairs are merged to create bigger clusters. Agglomerative hierarchical clustering is primarily used for power consumption monitoring [16,17] and load balancing [18]. Given that each device is independent and can function autonomously, we will adopt the agglomerative hierarchical clustering in our approach.

### 3.2. Device Clustering

In the context of device spatial clustering, researchers adopted different methodologies and use cases.

The authors in [19] presented different constraints that affect the result of a clustering algorithm. The center and the capacity of each cluster and outliers must be taken into consideration while clustering IoT devices. Non-spatial attributes may also interfere with the clustering process. The authors also proposed an algorithm that covers these constraints. They created an objective function that relies on distance, centers, and outliers. The algorithm is composed of two main steps. The allocation step assigns each point to the nearest facility (center), and the location step relocates centers following the newly assigned points.

In Ref. [20], the authors elaborated an algorithm that clusters IoT devices while taking into consideration obstacles. The algorithm starts with m cells. Each cell is denoted as dense/non-dense and obstructed/non-obstructed. Obstructed cells are cells having obstacles, while dense cells are cells having a lot of devices. Neighboring dense and non-obstructed cells are merged into a single cluster. The output of the algorithm returns a list of clusters along with their centers.

In Ref. [21], the authors took the case of device spatial clustering in catastrophic disasters. To gather information, a UAV is deployed over the desired area. The UAV gathers info from cluster heads. Cluster heads are responsible for gathering the data from cluster members and transmitting them to the UAV. Cluster heads are chosen based on the energy levels of the cluster nodes.

Energy and correlation principles are employed in the spatial device indexing algorithm presented in [22], extending the network life cycle. First, the nodes are divided into two clusters following their energy consumption. After dividing the clusters, upon receiving a task, a node acts as an initiator for the task. After receiving the information from the different nodes, the initiator compares the amount of information from the old cluster with the newly formed cluster. If the amount of information is greater than the old cluster, the join request (task) is accepted and the cluster is created.

In Ref. [23], a user-centric clustering approach is given. The authors showed different constraints faced during a user-centric clustering process, such as traffic load, security, delay, mobility, energy management, and computational capability. In the proposed algorithm, the network architecture is composed of access points, user equipment, and a macro base station. The connection across access points and user equipment is determined by their distance, which cannot exceed a predetermined limit. The AP is all in the range of the macro base station.

In Ref. [24], the authors proposed an algorithm to cluster devices following their energy and their distance to the base station. Nodes having minimal residual energy and being closer to the sink are elected as CH (cluster head). In addition, nodes having an optimal level of energy are elected as active nodes for an area. Messages are exchanged between the CH and the sink using a multi-hop communication.

In Ref. [25], an energy optimization method is proposed where the authors presented a socially aware clustering technique. Using this technique, a device receiving information from many devices is elected as CH. Cluster heads can have many cooperators in order to transfer the message to a sink node. The network can have many sinks; thus, the CH will forward the message through coordinators, reducing the energy consumption of the devices and increasing the network lifetime. Device storage capabilities are not mentioned in their work, plus the authors grouped the devices following their energy consumption without taking into consideration their coverage area.

In Refs. [26,27], the authors proposed a clustering technique that involves cluster heads and subcluster heads (SCH). Cluster heads are chosen based on the resource capability of a node precisely following their residual energy, computational capability, and storage capacity. Cluster heads and subcluster heads perform aggregation operations, reducing traffic load on the network, hence extending its lifetime. They also proposed an architecture named CCIC-WSN, extending the NDN (named data networking) architecture, where communication packets of CH and CM and their tasks (data aggregation and management tasks) are detailed. To optimize data retrieval, a lite-query structure is proposed, allowing for filtering the content based on dynamic keywords and comparison operators.

Device clustering is also popular in fog computing approaches. For example, in [28], the authors presented an approach named I-SEMP. Devices can choose to communicate between a fog node and another device that operates as a small proxy (SP) in case fog nodes (FN) are far from the current device. These choices are made following the energy consumption, the residual energies in the fog network, and the distances between the involved nodes. After choosing the SP, the different nodes can decide to be connected to an SP or an FN, depending on the distance between these nodes. Following the energy consumption of the devices, new devices can be elected as an SP at the end of each round (iteration). The experiments in this approach showed good results compared with other approaches.

### 3.3. Agglomerative Hierarchical Clustering

Agglomerative hierarchical clustering (AHC) is the grouping of many singleton clusters in order to create bigger clusters incorporating leaf nodes. AHC was used by many researchers a long time ago, such as in [29]. The authors presented a protocol-based clustering technique implying a cost function. The cost function must be minimized in a way that an optimal number of clusters is reached starting from leaf nodes. The cost function has two main elements: the first element controls the cluster shape and size, while the second element controls the cluster size. The use of these elements will lead to have an optimal number of data clusters according to the cost function.

In modern approaches, AHC computes a distance matrix between the involved nodes using criteria referred to as linkage. The most recognized linkage types are single-linkage clustering, where the minimal distance between individuals is considered; complete-linkage clustering, where the maximal distance between individuals is employed; and average linkage clustering, where the average distance between individuals is used. For example, in [16], the authors used the dynamic time warping distance as a distance metric, replacing the traditional Euclidean distance method used in many clustering techniques. The usage of the dynamic time warping distance metric is effective in clustering time series data points. The authors evaluated the clustering performance with many distance metrics and a linkage. Following the experiments conducted, AHC clustering with a DTW distance metric with a complete linkage gave the best results for time series.

### 3.4. Comparison Table

In Table 1, we compared the aforementioned approaches with three criteria: coverage range (i.e., the area in which the device can sense information about a specific attribute such as temperature or humidity), energy/power, and capacity (mainly the storage). Columns marked with a check mark indicate that the approach considered the corresponding criterion, while those marked with a “-” did not.

Only the two approaches provided in [20,24] considered in their clustering the coverage range of the sensors connected to the devices. We note that some approaches, such as in [23], use the term “Coverage range” to designate the communication range of the device. Since our criterion is the sensing area, an “-” is marked for this record. Regarding the energy criterion, many approaches (e.g., [21,22,23,24,26,27,28]) considered it for their clustering algorithms since the power is a major factor. The last column indicates that only [19,26,27,28] took the device capacity into consideration.

To sum up, one can easily see that none of the existing approaches consider all of the three criteria as does our approach.

## 4. Preliminaries and Assumptions

In this section, we aim to define the terminology employed and the assumptions of this proposal.

### 4.1. Device, Sensor, and Cluster Head

**Definition** **1**(Device). *A device d, also named IoT resource, can be defined by a 6-tuple as follows:*
(1)d:〈id,l,c,p,S〉*where*
*id is the device identifier;**l is the device location stamp (see Remark 1);**c is the device storage capacity (in bytes);**p is the device current power level (in Wh); and**$S={⋃}_{i=0}^{n}s$ is the set of the device sensors. Each sensor is defined as s:〈o,cz,ch〉, where*-*o is an observation (i.e., sensed data such as temperature);*-*cz is its coverage zone (see Definition 3); and*-*ch: is its cluster head (identifier) when it exists.▪*

It is to be noted that each device can identify its neighbors within the network. Network discovery protocols such as the Constrained Application Protocol (CoAP) and Simple Service Discovery Protocol (SSDP) enable devices and resources to manage and communicate with their neighbors and other entities.

**Remark** **1.**
*We note that the definitions of a location stamp and of an observation are mentioned in our previous work [30].*


**Definition** **2**(Cluster head). *A cluster head ch is a device (that inherits the attributes of the device object) responsible, regarding one or several observations, for providing functionalities and services for its cluster members such as data aggregation and load balancing. It is defined as follows:*
(2)ch:〈d,D,cz〉where:
*d = is the corresponding device**$D={⋃}_{i=0}^{n}s$: is the set of devices managed by the device.**cz: is the covered zone of the entire devices. ▪*

### 4.2. Zones and Environment

**Definition** **3**(Zone). *A zone z is a surface area represented by*
(3)z:〈id,su,shape,L〉where
*id is the zone identifier;**su is the surface of the zone;**shape is the spatial shape of the zone; and**L:$={⋃}_{i=0}^{n}l_{i}$∀i∈N is the set of location stamps that constitute the vertices of the zone. ▪*

**Remark** **2.**
*A covered zone, denoted by cz, is a zone that is covered by at least one device. A cz can have many vertices. The covered zones of the devices are spatially sampled into rectangles/squares, simplifying the calculations during the clustering. This step is explained in detail in the upcoming sections.*


**Remark** **3.**
*An uncovered zone, denoted by uz, is a zone that has no device inside, thus having no coverage. A uz has two vertices only. Uncovered zones are always represented in our study by rectangles, leading to the necessity of two vertices to represent the zone.*


**Definition** **4**(Environment). *An environment is an area that groups the set of covered and uncovered zones. It is represented by a rectangle and is defined as*
(4)env:〈id,CH,UZ,L〉where
*id is the environment identifier;**CH is the set of cluster heads in the environment;**UZ is the set of uncovered zones in the environment; and**L are the two corresponding vertices of the environment. ▪*

## 5. Proposed Approach

In this section, we present the global architecture used in our approach.

Our approach aims to cluster devices according to their coverage range, capacity, and power. As shown in Figure 2, the clustering protocol takes the deployed devices as an input. It is to be noted that device deployment is assumed in our study to be random because, in many situations and use cases, it is not always controlled.

Once the devices are deployed, the clustering protocol begins processing this input. The input consists of a set of devices, and the output is the clusters of devices along with a set of uncovered zones. Our clustering protocol is divided into three layers:

### 5.1. Pre-Clustering

In this layer, there are three main steps that are required to optimize the protocol performance:-Environment division following physical communication: commonly, devices are grouped based on their physical communication connections. In other words, each group of devices, which can communicate with each other using direct or multi-hop links, is gathered together. This ensures an overlay connection between the devices (where the algorithm execution must occur). IoT communication technologies (such as Wi-Fi and Bluetooth) enable direct communication, allowing nearby IoT devices to easily interact, thereby simplifying the task of this module. In Ref. [31], the authors considered the communication range as two times the coverage range. Other approaches, such as in [32], rely on connecting the devices from the beginning; hence, each device knows its directly connected devices. In our approach, we assume that each group of devices is aware of its directly or indirectly connected neighbors (using any network discovery protocol), which allows smooth inner connection.-Coverage area conversion: following [31], sensing models are the representation of sensing capabilities and quality. They rely on the sensing method: (1) directional sensing that depends on the distance and the horizontal orientation of the sensor or (2) omnidirectional sensing that refers to devices that can capture a 360-degree view of the surrounding scene (i.e., determines if a point is within the sensor’s radius). Several sensing models can be distinguished, but mainly two: Boolean and probabilistic. The Boolean sensing model is one of the most used according to [31]. It consists of considering each sensor node to have a binary sensing capability within a specific radius; i.e., it can detect the presence or absence of a target. The probabilistic sensing model extends the Boolean sensing model to better reflect modern connected environments. It relies on the probability of detecting a point within the coverage area. Even if a point is within this area, the detected value might have low accuracy or be undetectable. In our approach, the probabilistic sensing model is adopted since it includes the Boolean and provides more realism. In this study, we only focused on omnidirectional sensing. To reduce computation complexity in our approach, we transform the devices’ coverage zones (circles and lines) to either squares or rectangles and incorporate a probabilistic percentage named degradation percentage based on the sensor specifications and environmental conditions.      To ease the illustration of the coverage, let us consider Figure 3. There are two ways to represent an omnidirectional device coverage: one method uses a square with a side equal to 2∗R (case (a)), and the other represents the side of the square by $R*\sqrt{2}$ (case (b)). In case (a), some of the coverage area exceeds the coverage range of the sensor (gaining a small portion from the coverage range), while in case (b), the coverage area is totally included in the coverage range while losing a small portion of the coverage range. The degradation percentage (DP) directly impacts the coverage range, as demonstrated in cases (c) and (d). As the DP goes up, the coverage range area will be shortened. The method for representing the coverage area with the DP depends on the sensor specifications and precision.-Device grouping: After generating the coverage zones of each device, we check for continuous intersections between them. Devices having a continuous intersection will be added to the same group. At the end of this step, the devices can communicate with each other and have consecutive covered zone intersections. The clustering algorithm will be applied independently to each group.

### 5.2. Clustering Algorithm Execution

In this layer, the clustering algorithm is executed along with some pre-processing steps (commonly performed in many clustering algorithms).

-Sensor metadata normalization: all numerical values are normalized using the following min–max normalization technique [33]:
(5)$$N\left(x\right)=\frac{x-x_{min}}{x_{max}-x_{min}}$$
with $x_{min}=0$ and $x_{max}$ the highest possible value. *x* represents any numerical value between $x_{min}$ and $x_{max}$.-Coverage zone clustering algorithm execution: the clustering algorithm is executed for each device group. Details about the execution steps will be presented in Section 6. The clustering process will result in distinct, non-overlapping covered zones, each with a designated cluster head. It is important to note that any grouping, division, or modification of a device’s coverage area will create a covered zone.

### 5.3. Post-Clustering

In this layer, we gather additional information from the environment that could be used to simplify other networking tasks. These steps are optional but ought to enhance many networking tasks (device indexing and information gathering).

-Uncovered area calculation using ENV and CZ: In this step, we calculate empty areas by subtracting the entire connected environment from the covered zones. As a result, we will have areas that are not covered by any sensing device.-Uncovered zone division using internal vertices: Internal vertices are those located within the boundaries of the connected environment but not on the edges of the environment’s Minimum Bounding Rectangle. For each internal vertex, we draw a horizontal line dividing the current uncovered area into two parts. After splitting all internal vertices, the uncovered zones will be rectangular. This step aims to reduce the storage capacity required on the device’s local storage, as only two points are needed to represent a rectangle in memory. More information about this part will be given in Section 7.

## 6. RDSC Clustering Algorithm

In this section, we will detail the covered zone clustering algorithm. We will start by presenting the different equations employed. Then, we will present the different use cases. Finally, we will detail each step of the RDSC clustering algorithm.

### 6.1. Equations and Applications

#### 6.1.1. Objective Function

As previously discussed, three main criteria must be considered when clustering: (1) the surface area of the current cluster, (2) the power within the cluster, and (3) the vertices required to store the boundaries of the current cluster (depending on the device storage capacities). In our case, we aim to maximize the cluster’s surface area (reducing the overall number of clusters) and to increase the power contained inside the cluster, while minimizing the number of vertices to be stored to track the cluster boundaries. To state the problem, we defined an objective function that incorporates the above criteria. As shown in Equation (Equation 6), we can distinguish a gain (G) whose values are to be maximized and a loss (L) to be minimized.
(6)f(x)=G−L

In other words, the adopted problems are converted into a maximization problem, where we aim to maximize the gain of *f* and minimize the loss of *f*.
(7)$$f\left(x\right)=w_{1}*ch\left(x\right).cz.su+w_{2}*sum\left(ch\left(x\right).D.p+ch\left(x\right).p\right)-w_{3}*length\left(ch\left(x\right).cz.L\right)$$
(8)$$\left\{\begin{array}{c} w_{1}+w_{2}+w_{3}=1\\ 0\;\leq \;w\;\leq \;1 \end{array}\right.$$

In Equation (Equation 7), we detail the gain and the loss components of the objective function. $w_{1}$, $w_{2}$, and $w_{3}$ are the corresponding weights (provided manually) that affect the importance of the associated criteria. $w_{1}$ is associated with the surface of the zone cz.su. A higher $w_{1}$ will increase the importance of larger surfaces for the objective function. $w_{2}$ is associated with the power of all the devices combined including the power of the cluster head ch. Given that energy can be quantified in watt-hours (Wh), the energy of the covered zone is calculated as the sum of the power of each device within that zone. A higher value for $w_{2}$ will amplify the significance of increasing the power levels within the objective function. $w_{3}$ is associated with the number of location stamps required to store the zone cz.L. During the algorithm iterations, a custom-shaped zone will have a variable number of vertices. This will necessitate a greater storage capacity on the device to accommodate the information.

#### 6.1.2. Use Cases

To apply the objection function, a comparison between two intersecting covered zones must be conducted. Since our coverage zones are represented by rectangles and squares, we can identify five main use cases. These use cases are illustrated in Figure 4. Figure 4a,b are two different zones each zone having a separate color.

In Figure 4 case (1), cz (a) remains intact while cz (b) is divided, resulting in cz (b) having six vertices. In case (2), cz (a) is divided while cz (b) retains its shape, resulting in cz (a) having six vertices. cz (a) and cz (b) can be merged into a single covered zone as shown in case (3) having eight vertices. In case (4), cz (a) is shrunk, reducing its surface while removing its intersection from zone b. In case (5), cz (b) is shrunk and its intersection with cz (a) is removed. The option to shrink zones is not always considered since, in some connected environments, this is not an option. In other situations, shrinking cz can improve network lifetime, for example, by reducing the battery-drained devices’ impact on the network. Using the objective function (Equation (Equation 7)), each use case will have specific values based on the surface area, vertices, and total power of each separated, merged, or shrunk zone.

For use cases (1) and (2), the final objective value can be determined by summing the values of each zone alone. We note that, in these section’s equations, a and b are the zones represented in Figure 4. After applying the zone dominance, we will obtain two zones. One unchanged zone has its objective value. Another zone has been cut, reducing its surface and increasing its vertices. To obtain a single value for these zones, we added their objective function values. Using Equation (Equation 7), we can deduce
(9)f(a|b)=f(a)+f(b)

For use case (3), the final objective value is the objective value of the two combined zones. It should be noted that the objective value for combined zones differs from the objective value of two separate zones that are divided. After merging (unifying) zones, we can easily calculate their combined objective value. To calculate the final objective value for two merged cz, we define the following equation:(10)f(a+b)=f(a∪b)

For use cases (4) and (5), the final objective value is the value of the zone that we kept minus the value of the affected zone multiplied by a deletion rate (loss rate). Shrinking zones signify that we are losing data. Data loss should be penalized depending on the environment and user needs. The deletion rate is a value passed as an input at the beginning of the algorithm that affects the final result of the objective function. Recall that device exclusion is optional. It is not required depending on user needs. Having two zones, a and b, we subtract the loss of zone b (f(b)∗deletion_rate) from zone a. The final objective value associated with cz narrowing is defined by
(11)f(a−b)=f(a)−f(b)∗deletion_rate

After calculating the values resulting from Equations (Equation 9)–(Equation 11), we compare the different values and we choose the case that have the highest objective value. The according value will have the highest priority to be executed.

Algorithm 1 shows the execution and calculations involved in each use case. The inputs of the algorithm are the two cluster heads, the weights of the surface area, the power, and the number of vertices, respectively. The merge factor and the deletion rate are inputs to pass to further processing steps in the algorithm. The output of the algorithm is the calculated value for each use case with corresponding metadata.
**Algorithm 1:** runUseCases()**Input**:   $ch_{1},ch_{2},w_{1},w_{2},w_{3},mergeF,deletionR$**Output**:   useCasesObj        //object containing use cases results**Local Variables**: useCasesObj=[]//Calculating power values**1**$powerch_{1}=ch_{1}.p+sum\left(ch_{1}.D.p\right)$**2**$powerch_{2}=ch_{2}.p+sum\left(ch_{2}.D.p\right)$**3**resultDefaultZone1 = $objectiveFunction(ch_{1}.cz.su,powerch_{1},length\left(ch_{1}.cz.L\right),w_{1},w_{2},w_{3})$**4**resultDefaultZone2 = $objectiveFunction(ch_{2}.cz.su,powerch_{2},length\left(ch_{2}.cz.L\right),w_{1},w_{2},w_{3})$//merged use case**5**useCasesObjElement.label = ‘merged’**6**useCasesObjElement.cz = $combineCoverageZone(ch_{1}.cz,ch_{2}.cz)$**7**useCasesObjElement.P = $powerch_{1}+powerch_{2}$**8**useCasesObjElement.mergedValue = $mergeF*(length\left(ch_{1}.D\right)+\left(ch_{2}.D\right)+2)$**9**useCasesObjElement.value = $objectiveFunction(useCasesObjElement.cz.su,useCasesObjElement.P,length\left(useCasesObjElement.cz.L\right),w_{1},w_{2},w_{3})$**10**useCasesObj.push(useCasesObjElement)//dominant one use case**11**useCasesObjElement.label = ‘dominantOne’**12**useCasesObjElement.cz = $ch_{2}.cz$−$ch_{1}.cz$**13**resultZone2 = $objectiveFunction(useCasesObjElement.cz.su,powerch_{2},length\left(useCasesObjElement.cz.L\right),w_{1},w_{2},w_{3})$**14**useCasesObjElement.value = resultDefaultZone1 + resultZone2**15**useCasesObj.push(useCasesObjElement)//dominant zone two use case**16**useCasesObjElement.label = ‘dominantTwo’**17**useCasesObjElement.cz = $ch_{1}.cz$−$ch_{2}.cz$**18**resultZone1 = $objectiveFunction(useCasesObjElement.cz.su,powerch_{1},length\left(useCasesObjElement.cz.L\right),w_{1},w_{2},w_{3})$**19**useCasesObjElement.value = resultZone1 + resultDefaultZone2**20**useCasesObj.push(useCasesObjElement)//shrink zone one use case**21**useCasesObjElement.label = ‘shrinkOne’**22**useCasesObjElement.cz = $ch_{2}.cz$−$ch_{1}.cz$**23**useCasesObjElement.value = resultDefaultZone1−resultDefaultZone2 * deletionR**24**useCasesObj.push(useCasesObjElement)//shrink zone two use case**25**useCasesObjElement.label = ‘shrinkTwo’**26**useCasesObjElement.cz = $ch_{1}.cz$−$ch_{2}.cz$**27**useCasesObjElement.value = resultDefaultZone1−resultDefaultZone2 * deletionR**28**useCasesObj.push(useCasesObjElement)**29**return useCasesObj

   In lines 1–2, we calculate the power contained in each cluster. The power is measured in watt-hours (Wh), leading to the summation of the power values determining the total power in the according cluster. Default objective values are also calculated for each cluster head (lines 3–4). These values will be used in the upcoming steps.

After calculating the required values engaged in further steps (the power and the default objective values), we start calculating the objective value of each use case. First, the algorithm starts with the merged use case. In the merged cz use case, the cz of both cluster heads are combined, creating a unique zone. The combination of these zones will determine the total surface area and the number of vertices needed to store the boundaries of the cz. The power of the two cluster heads is added. The result of combining both powers determines the total power inside the newly merged zone (lines 6–7). The mergedValue in line 8 is calculated by multiplying the passed merge factor with the total number of devices in the current group. In lines 9–10, the objective value calculation for the merged cz is performed. The obtained values are added to the useCasesObj.

The next use case concerns the domination of $ch_{1}$ over $ch_{2}$. During this step, $ch_{1}$ remains the same, while changes affect $ch_{2}$ only. The new cz of $ch_{2}$ will be its current cz subtracted by the cz of $ch_{1}$ (line 12). The purpose of this step is to remove the intersecting area between $ch_{1}$ and $ch_{2}$ from the cz of $ch_{2}$. The new objective value of $ch_{2}$ is calculated in line 13, while the objective value of $ch_{1}$ remains the same (no changes are made to $ch_{1}$). Both objective values are summed and pushed to the useCasesObj (lines 14–15).

Similar to the dominant first use case, the $ch_{2}$ cluster dominates $ch_{1}$ in the dominant second use case. The intersection area is assigned to $ch_{2}$ while being removed from $ch_{1}$ (line 17). In lines 18–20, objective values for $ch_{1}$ and $ch_{2}$ are calculated separately, summed, and pushed to the useCasesObj.

For zone shrinkage use cases, the intersection area between the cz is removed. Then, we subtract the default objective value of the shrunk zone multiplied by the deletion rate deletionR from the objective value of the current zones, and we add their values to the useCasesObj array (line 21–28).

Finally, we return the useCasesObj array used in Algorithm 2.
**Algorithm 2:** spatialClustering()
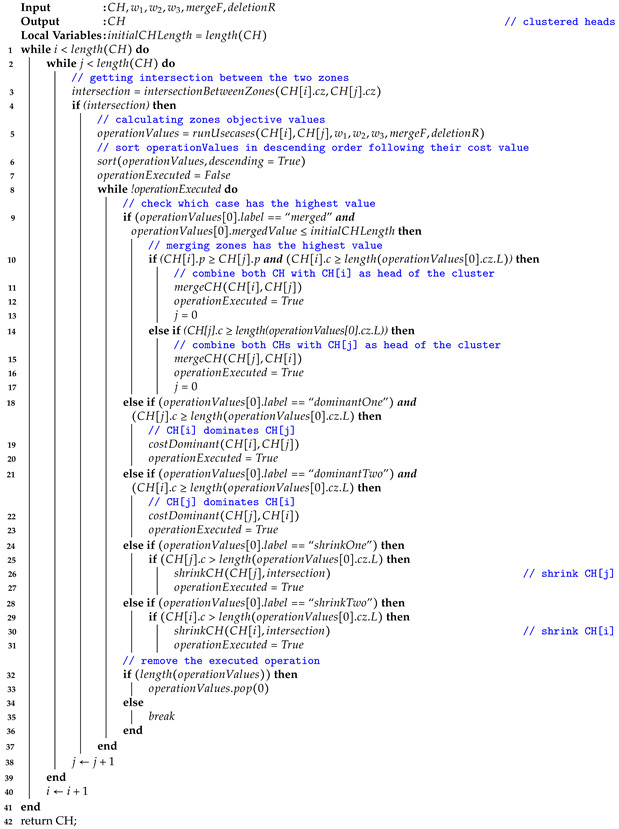


### 6.2. Main Algorithm Execution

After completing the use case calculations, we start applying changes to the corresponding cluster heads CH. The changes to CH will be explained in detail in this section.

The algorithm flow is demonstrated in Figure 5. The algorithm starts with two loops going through all the CHs’ detecting intersections between those zones (lines 1–3). When an intersection is detected, we can start applying the different use cases of Algorithm 1 (line 5). Following the score obtained by the runUseCases function, we can mutate our CH. The results are sorted in descending order, placing the highest value at the beginning (line 6).

When merged zones have the highest values, we check if the mergedValue is less than the total number of devices in the environment. This step regulates the device density within the clusters, preventing the formation of overly large clusters (line 9). After checking the cluster density allowance, we identify the cluster head with the highest power between the two. The purpose of this check is to enhance network health by designating high-power devices as a cluster head. As mentioned before, our clustering algorithm considers the device capacities while clustering. In addition to the power condition, we check for the capability of the device to store all the boundaries of the new covered zone cz.L. The device with the highest power capable of storing the zone will be chosen as cluster head ch. After choosing the ch, the cz are combined and the attributes of the designated ch are updated. After the update of the ch, *j* is set to 0 since the current zone merge can affect bypassed zones by creating new intersections with them. These steps can be seen in lines 9 to 17.

In case the first cluster head dominates the second, the new cz of ch must be checked to ensure that $ch_{2}.c$ supports storing all the vertices of the zone. Once checked, the overlapping zone is assigned to $ch_{1}$ (no difference for $ch_{1}$) and the new cut zone is assigned to $ch_{2}$. Lines 18 to 20 demonstrate this step.

The same conditions as in the cost dominant zone first use case are applied to the cost dominant zone second use case, but for $ch_{1}$. The algorithm checks that $ch_{1}$ can store all the vertices of the cut zone. The overlapping zone is assigned to $ch_{2}$ (lines 21–23).

The last two steps involve CH shrinking. Based on the calculations made, we can choose to shrink a device cz or exclude it from the algorithm, reducing its impact on the environment. Similarly, a check on the device storage capacity is made to check if the devices can store the cut zone. These steps are outlined in lines 24–31. From lines 32–36, these steps involve removing operations that could not be executed and proceeding to the next operations.

## 7. Uncovered Zones Division

As previously mentioned, generating uncovered zones is an optional step but can be beneficial for future network operations. Having additional information about the environment can assist in indexing, querying, and other networking tasks. For example, when querying the value of an attribute from a specific location stamp *l*, data retrieval can be more effective when knowing the boundaries of uncovered zones. In our case, uncovered zones are always represented as rectangles, which reduces their storage cost when they need to be stored on the device.

The input of the algorithm is the resulting output of Algorithm 2. The output of Algorithm 3 will be the environment containing the cz, uz, and the boundaries of the entire environment represented by two location stamps. In line 1 of the algorithm, an environment variable is initiated and filled with the CH and the location stamps of the bottom-left and top-right corners of the environment. These corners are represented by the $x_{min}$, $x_{max}$, $y_{min}$, $y_{max}$ between all the cz represented in the entire CH group.
**Algorithm 3:** generateUncoveredZones() **Input**    :CH **Output**    :env         //  the environment object**1**  env = initiateEnvironment(CH)**2**  emptyAreas = getEnvrionmentCoordinates(env,CH)**3**  UZ=splitOnInternalVertices()**4**  env.UZ=UZ**5**  return env

In line 2, empty areas are calculated by subtracting all the covered zones of CH from the surface of the env’s rectangle. The created empty areas are custom-shaped. To create rectangles from these custom shapes, the splitOnInternalVertices() method is used. This method detects internal vertices, which are vertices located inside the boundaries of the original env’s rectangle. After detecting the internal vertices, a horizontal split is performed on each internal vertex, transforming the custom shape into a multi-rectangle representation. Each rectangle represents a single uz. An array of uz is returned, denoted by UZ. The env variable is updated and returned as a final result of our clustering algorithm in lines (4–5). Figure 6 shows an example of the execution of Algorithm 3. The clustering result that is displayed is the output of 20 devices clustered together, each cluster is represented by a different color. After finishing the clustering process, all empty areas of the clustering result are filled with rectangles that will be used in further networking operations. Recall that only rectangles are obtained since they can be stored easily using two points. Taking the bottom-right uncovered zones as an example, to store the entire polygon, 13 vertices are required. After dividing the polygon into rectangles, 10 vertices are required, hence reducing the storage cost of uncovered zones.

## 8. Experiments

In this section, we present the set of experiments achieved on a 12 GB RAM machine with an Intel(R) Xeon(R) CPU @ 2.20 GHz (Google Colab). The first set of experiments denotes the algorithm performance evaluation including weights and density factor evaluation. A comparison of RDSC with DBSCAN and k-means was also made on 20 and 100 devices. For data transmission, we consider that the devices have the required parameters to transfer the messages over Wi-Fi channels (frequency band equal to 2.4 GHz and transmission power equal to 20 dBm). We consider that the IoT devices use the CSMA/CA access method to avoid collisions.

The complexity of the algorithm is $n^{2}\log n$ due to the two external loops (each loop is counted as n), while *j* is being set to 0 to recheck for bypassed intersections between zones (since, after a merge, the number of zones is reduced, we multiplied $n^{2}$ by logn ).

### 8.1. Performance Evaluation

During these experiments, we varied the different algorithm weights and parameters to evaluate the impact of each parameter on the final result of RDSC.

#### 8.1.1. Effect of the Surface Weight $w_{1}$ on the Clustering Results

In this experiment, we evaluate the effect of the surface weight $w_{1}$ on the final result of RDSC. On the other hand, $w_{2}$ is set to 0 to remove the effect of the device power on the result, while mergeF and deletionR are fixed to 3.5 and 0.3, respectively (Table 2). A group of 100 devices is generated. Recall that a group contains devices that have consecutive intersections.

We varied the values of $w_{1}$ to 0.9, 0.75, and 0.5, while adapting $w_{3}$ to keep the addition of both weights equal to 1. In Figure 7, Figure 8, Figure 9, Figure 10, Figure 11 and Figure 12, we demonstrate graphs showing the effect of $w_{1}$ and $w_{3}$ on the surface and vertices. In Figure 7, most of the clusters have big surface areas. Figure 8 shows that even though the clusters have big surface areas, they also have high vertex numbers. This behavior is expected since $w_{1}$ has a big weight on the objective function. In addition, the number of clusters is equal to 22, meaning that a lot of merges occurred during the execution of the algorithm, increasing the cluster’s surfaces. Notice that zones 14 and 15 have surface areas equal to 0 while having 0 vertices. This indicates that these two devices are excluded from all clusters while minimizing their covered zone until having a negligible area due to the coverage zone shrinkage step.

In Figure 9 and Figure 10, $w_{1}$ is set to 0.75, while $w_{3}$ is set to 0.25. The surface area of the clusters decreased, while the number of vertices per zone was reduced. Since $w_{1}$ decreased and $w_{3}$ increased, the number of vertices per cz has a higher impact when merging. A higher number of clusters are obtained since vertices affect the zone merging process, hence increasing the number of clusters. We can observe that the average surface per cluster decreases and is distributed across several clusters.

In Figure 11 and Figure 12, $w_{1}$ and $w_{3}$ have both been set to 0.5. Since more weight has been assigned to $w_{3}$, fewer vertices are accepted per zone, resulting in a higher number of clusters.

Comparing the aforementioned results, one can conclude that as $w_{1}$ increases, the average surface per cluster increases as well, reducing the number of overall covered zones.

#### 8.1.2. Effect of the Power Weight $w_{2}$ on the Clustering Results

In this experiment, we evaluate the power in wh contained in each cluster while changing the values of $w_{2}$ and $w_{3}$. $w_{1}$ is set to 0, removing the effect of cz surfaces on the execution of RDSC. mergeF and deletionR are fixed at 3.5 and 0.3, respectively (Table 3).

During this experiment, we decreased the value of $w_{2}$ from 0.9 to 0.5 while adapting $w_{2}$ accordingly ($w_{2}$ + $w_{3}$ = 1). From Figure 13, Figure 14, Figure 15, Figure 16, Figure 17 and Figure 18, we demonstrate graphs showing the effect of $w_{1}$ and $w_{2}$ on the power in Wh and on the number of vertices.

In Figure 13, many clusters have a cluster power greater than 10 Wh. Covered zone 15 reaches 60 Wh while having over 25 vertices. The average vertex number is high in most zones due to the low value of $w_{3}$ (Figure 14). Due to the many decisions regarding merging zones, the number of clusters is reduced from 100 singleton clusters to 22 clusters.

In Figure 15, the average power in each cluster drops since more zones having a power of less than 20 can be distinguished. Several zone vertices drop (compared with Figure 14) due to the higher importance of $w_{3}$ on the merging decisions.

Figure 17 and Figure 18 demonstrate a drop in the power of many clusters, while a decrease in the vertices number is visible. We can also distinguish a high increase in the number of clusters since it reaches 51 clusters, indicating a big number of singleton and duo clusters.

These graphs show that a higher value of $w_{2}$ will lead to clusters having high power capacities, while increasing the value of $w_{3}$ will result in a higher number of clusters, each cluster having a smaller value for power and vertices.

#### 8.1.3. Merge Factor Impact

We used the same group of 100 devices generated previously to assess the effects of the merge factor on the algorithm. We fixed the values $w_{1}=0.4$, $w_{2}=0.3$, $w_{3}=0.3$, and deletionR=0.3 and varied the merge factor mergeF (Table 4). Figure 19 shows the density value compared with the maximum number of devices per cluster. From mergeF=3 to mergeF=11, the max device is constant to 9 devices. This number is not affected by the merge factor since the first value affected by the merge factor is 100/11≈9. Then, the max number of devices decreases from 9 to 4, while the merge factor increases from 11 to 20. In conclusion, the maximum number of devices per cluster decreases as the merge factor increases.

### 8.2. 1000 Device Execution

In this experiment, we executed RDSC on a group of 1000 devices (having consecutive intersections) with the following parameters:Surface weight: 0.4;Power weight: 0.4;Vertex weight: 0.2;Merge factor: 3.5; andDeletion rate: 0.3.

The devices have a varied capacity between 10 and 30; in other words, they can store a maximum of 30 vertices. Their range is between 2 and 8 u. Their power is between 1 and 10 Wh. The final result contains 273 clusters.

As Figure 20 shows, most of the clusters have a power between 10 and 40 Wh. Figure 21 illustrates that most of the clusters have a surface greater than 200 $u^{2}$. On the other hand, Figure 22 indicates that there are singleton clusters. This is due to the storage and power limitations of the devices. Last but not least, Figure 23 proves that no device stores a zone that has more than 30 vertices.

In this experiment, one can deduce that RDSC gives good results for large IoT networks since it takes the device coverage range, storage capacities, and power storage while clustering.

### 8.3. Algorithms Comparison

In this experiment, we compare the results of DBSCAN and k-means with the RDSC algorithm. We generated two different groups. The first group has 20 devices, while the second has 100 devices with consecutive intersections. The algorithms have been configured as shown in Table 5:

As shown in Figure 24, five clusters of devices were generated, each cluster having a cluster head that has the highest power resources and that can store all the boundaries of the zones.

In Figure 25, the same 100 devices deployed before were used. Numerous clusters were generated, each cluster having the device with the highest power as ch.

In Figure 26, DBSCAN was applied on the same example with 20 devices. As shown in the graph, DBSCAN grouped the devices without taking into account device capacities. No single device can store the metadata of the entire environment. When devices are grouped together into a single cluster, a ch must be aware of its cluster members, making it difficult to store metadata of many devices at the same time due to the capacity limitations of the devices. In addition, the intersection of sensing areas between clusters will lead to data redundancy and regression of the network’s performance.

In Figure 27, DBSCAN was applied to 100 devices. DBSCAN clusters the devices following their density. It groups points where high density is detected. For 100 devices, DBSCAN gave a better result than the case of 20 devices. However, the intersection between clusters will cause data redundancy in the network, which can reduce network availability. Device capacities are also a major problem in that case due to the limitations of IoT devices.

In Figure 28, k-means was applied *k* (equal to 5 here), which specifies the number of clusters that must be generated. For 20 devices, k-means gave approximately similar results to our algorithm except for the intersection between the clusters. Comparing k-means with RDSC, RDSC automatically gave five clusters, identifying the optimal number of clusters.

In Figure 29, for 100 devices and using the same configuration, k-means gave huge clusters since it is limited to five clusters only. Huge clusters are not optimal in connected environments since IoT devices cannot store big data on their physical memories.

In this experiment, RDSC showed that it is effective for small and large groups compared with other clustering techniques.

### 8.4. Discussion

In these experiments, we demonstrated that our approach clustered the devices, taking into account their storage capacity by reducing the number of vertices. It also partitioned their coverage range while removing intersection areas. Last but not least, the power was taken into consideration by choosing to minimize or maximize the power impact on the result. The experiments demonstrated that the weights assigned to the objective function have an important impact on the final result. Increasing the weights of the surface will lead to more merges occurring. The same for the power weight: increasing $w_{2}$ will increase cz combinations. The amplification of $w_{3}$ will encourage zone splitting. In other words, the gain part of the objective function *G* favors merges, while the loss part *L* leads to splits. The users must find an equilibrium between *G* and *L* that suits their connected environment needs. Finding this equilibrium depends on the device’s storage capacities, power capabilities, and coverage range. Meanwhile, this equilibrium can also vary depending on the user specifications. The users can choose to focus on surface/vertex criteria, neglecting the power factors. RDSC gave good results compared with other algorithms, especially in the determination of the number of clusters automatically. Moreover, the experiments indicated that cluster overlapping is considered in RDSC, while other clustering techniques do not consider coverage zone overlapping between cluster heads. Following these experiments, the parameters must be chosen wisely, which can be challenging in some cases. In our approach, the parameters are very important. The number of devices inside the cluster, their coverage range, and the weights have significant effects on the final result. For example, by choosing the correct parameters, we can reduce the power consumption of the devices by choosing the correct power weight, merge factor, and deletion rates. For other environments, by choosing the correct parameters, the coverage range can be greatly enhanced compared with the number of devices. In other cases, a better device distribution can be achieved by varying the merge factor. In other words, the parameters must be adapted to the user’s needs. Further experiments will be conducted to determine parameter recommendations, depending on user needs and environmental conditions. In addition, connectivity must be taken into consideration to enhance the network performance.

## 9. Conclusions and Future Works

In this paper, we presented an approach named Range-Based Device Spatial Clustering for IoT networks (RDSC). The increase in IoT resources necessitates technologies that group sensory devices, helping in further network organization and operations. Device grouping leads to better network scalability, load balancing, data aggregation, and energy optimization. Device heterogeneity makes device grouping more challenging, especially due to the limited storage capacity and power of IoT devices. An overview of clustering algorithms was made while exploring device clustering use cases focusing on the device coverage ranges’ heterogeneity, energy capabilities, and device storage capacities. The definitions of the main components of the algorithm were demonstrated while explaining a new clustering approach that groups the devices into non-overlapping clusters considering the devices’ coverage ranges, storage capacities, and energy levels. Moreover, a network partitioning was performed on non-covered areas, gathering additional network metadata that can be used in further networking operations. Intensive experiments on the algorithm were executed and gave good results. A comparison of the RDSC technique with other clustering algorithms was performed, highlighting the varying results produced by these methods. Parameter recommendation techniques can be implemented in future works to help users to find appropriate configurations depending on their current needs and environmental conditions. Moreover, as a future work, we will consider the connectivity to enhance the network performance.

## Figures and Tables

**Figure 1 sensors-24-05851-f001:**
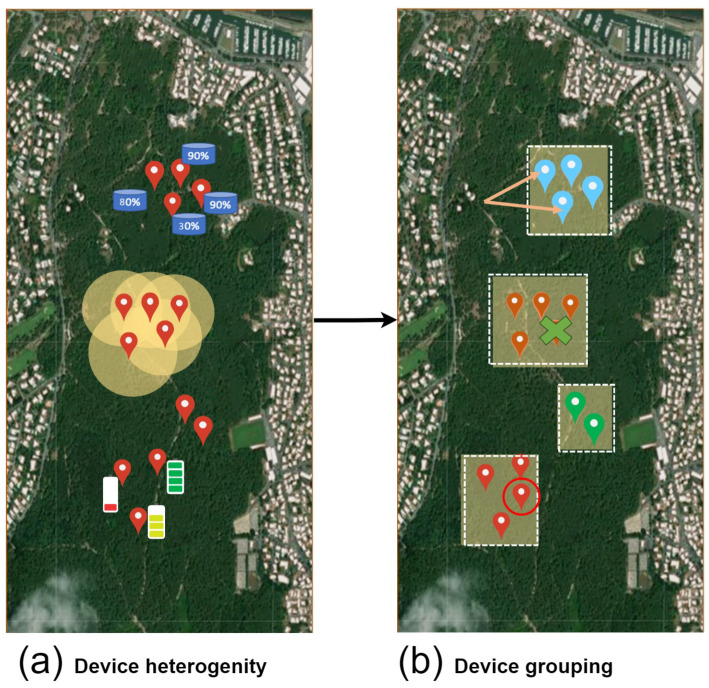
Deployment of devices in the Chiberta forest.

**Figure 2 sensors-24-05851-f002:**
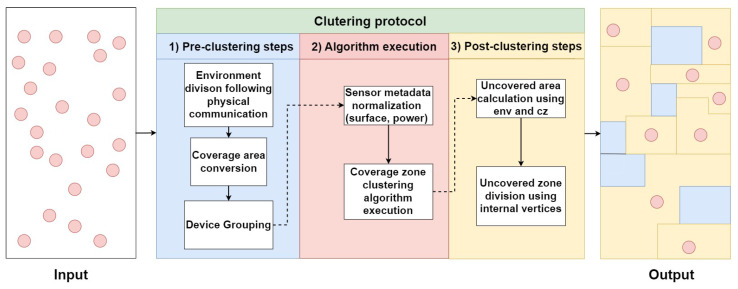
Our approach architecture.

**Figure 3 sensors-24-05851-f003:**
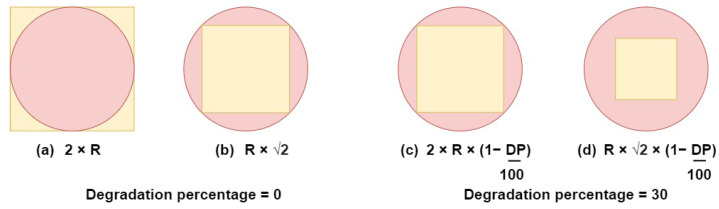
Coverage range transformation.

**Figure 4 sensors-24-05851-f004:**
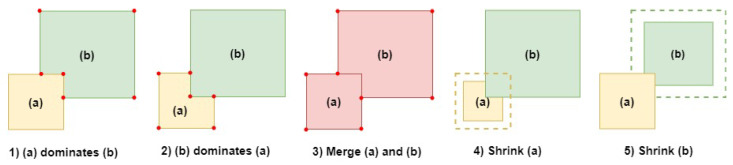
Objective function application use cases.

**Figure 5 sensors-24-05851-f005:**
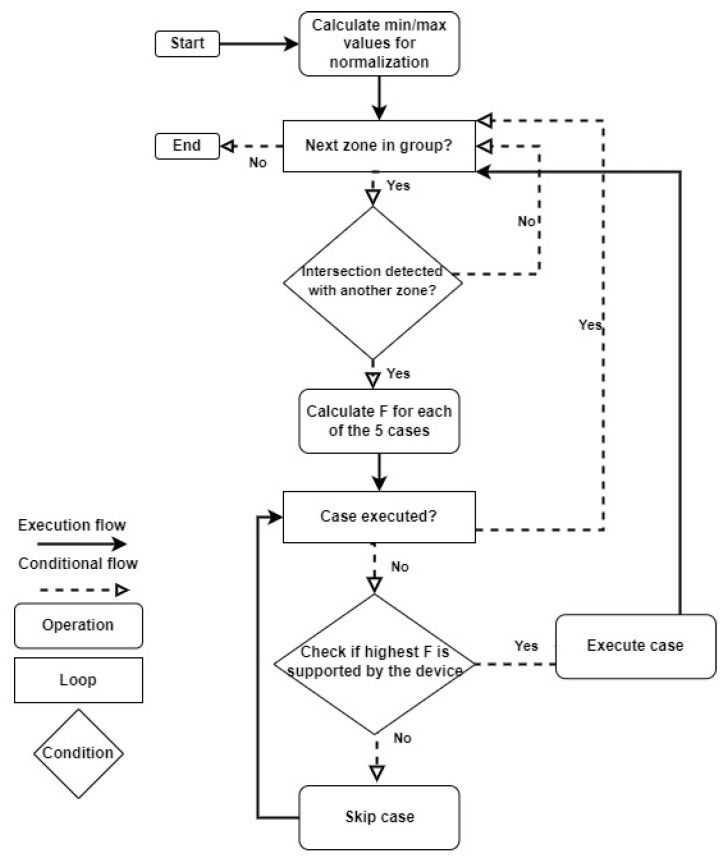
Clustering algorithm schema.

**Figure 6 sensors-24-05851-f006:**
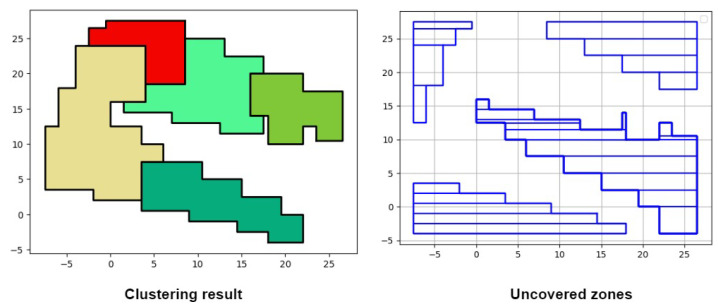
Uncovered zone example.

**Figure 7 sensors-24-05851-f007:**
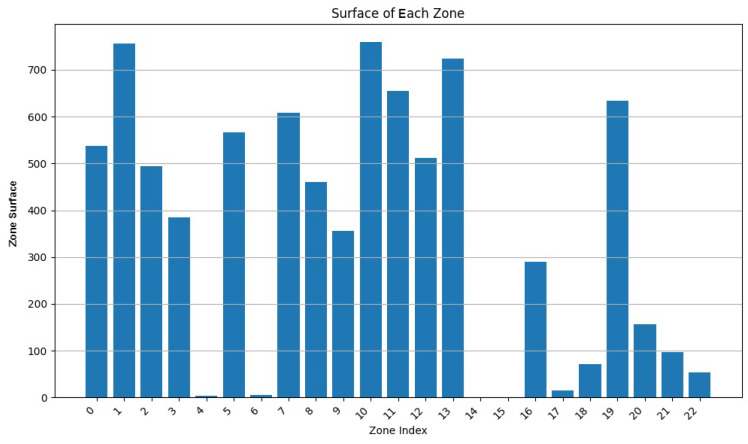
Cluster surface area for $w_{1}=0.9$ and $w_{3}=0.1$.

**Figure 8 sensors-24-05851-f008:**
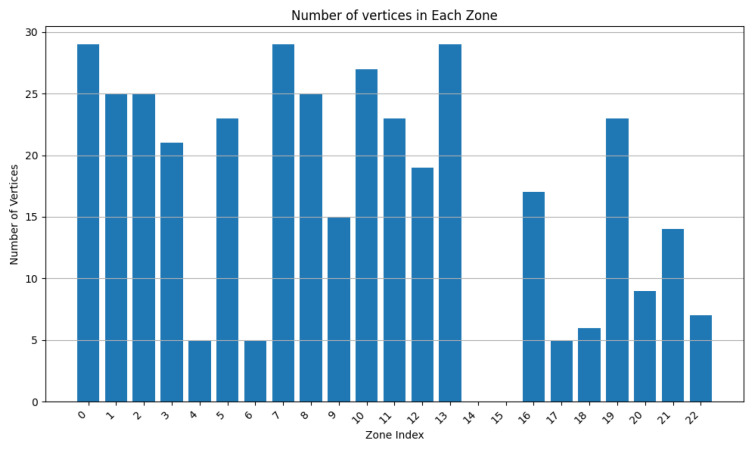
Cluster vertex number for $w_{1}=0.9$ and $w_{3}=0.1$.

**Figure 9 sensors-24-05851-f009:**
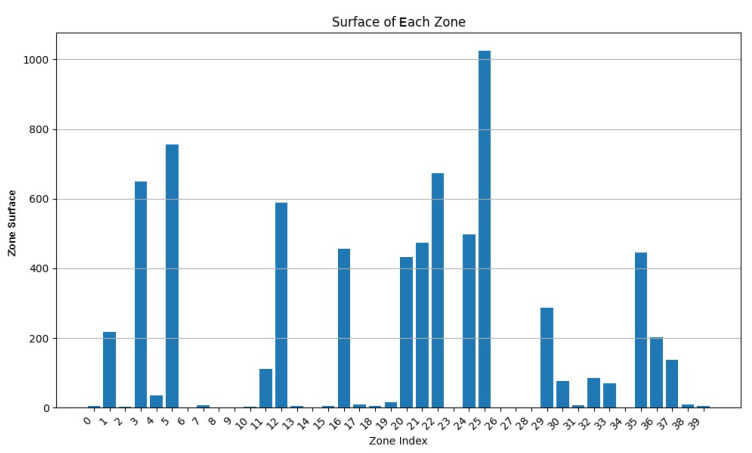
Clusters surface area for $w_{1}=0.75$ and $w_{3}=0.25$.

**Figure 10 sensors-24-05851-f010:**
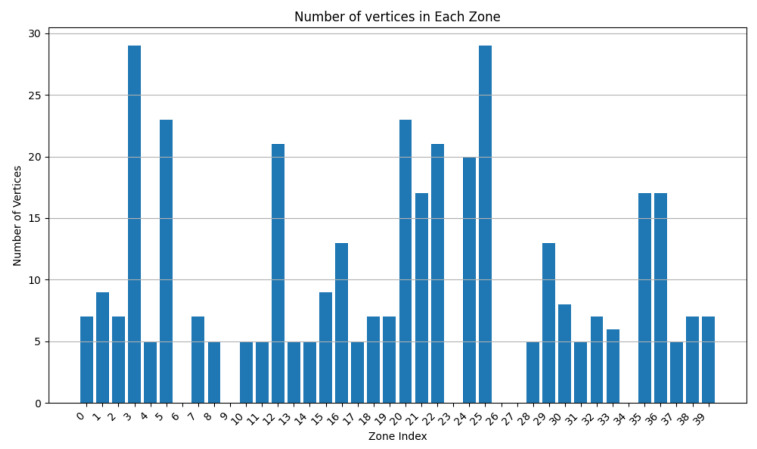
Cluster vertex number for $w_{1}=0.75$ and $w_{3}=0.25$.

**Figure 11 sensors-24-05851-f011:**
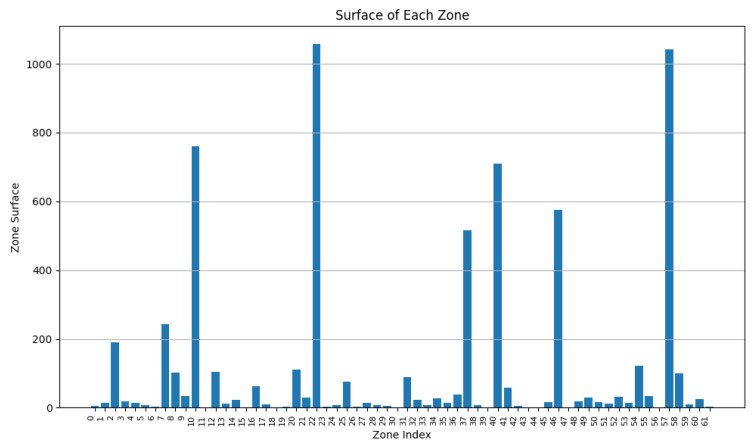
Cluster surface area for $w_{1}=0.5$ and $w_{3}=0.5$.

**Figure 12 sensors-24-05851-f012:**
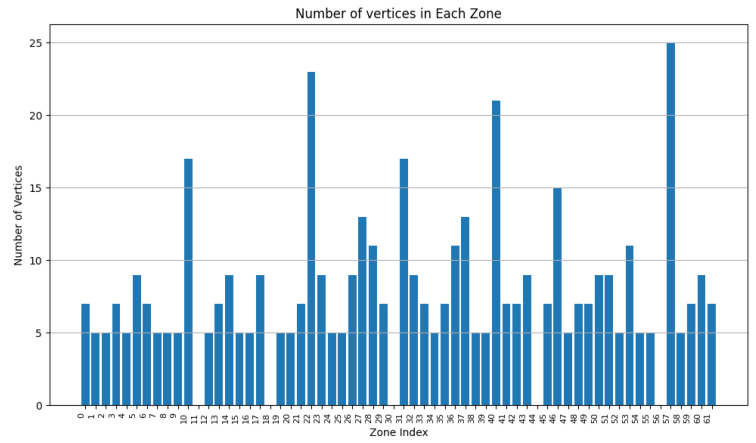
Cluster vertex number for $w_{1}=0.5$ and $w_{3}=0.5$.

**Figure 13 sensors-24-05851-f013:**
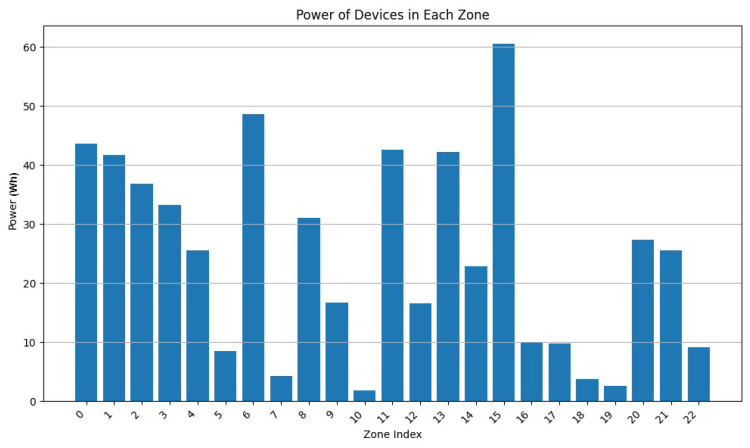
Cluster power for $w_{2}=0.9$ and $w_{3}=0.1$.

**Figure 14 sensors-24-05851-f014:**
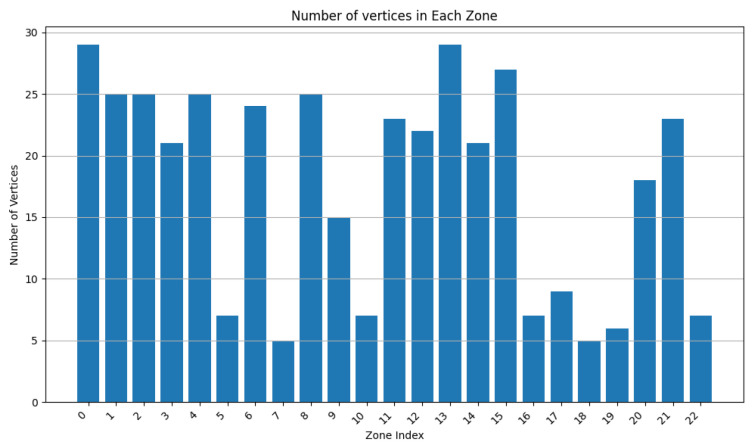
Cluster vertex number for $w_{2}=0.9$ and $w_{3}=0.1$.

**Figure 15 sensors-24-05851-f015:**
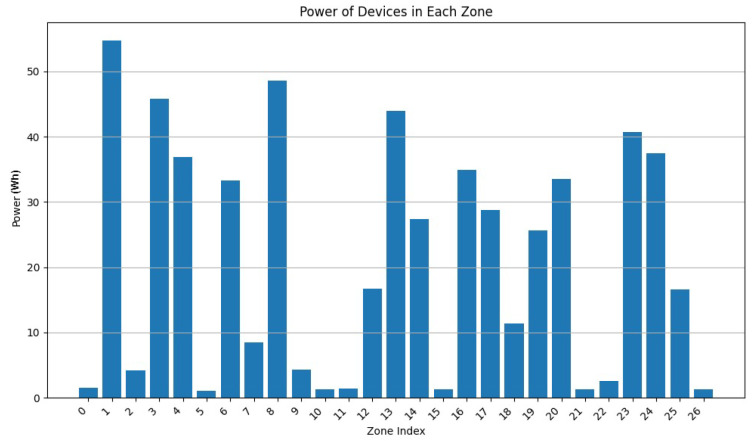
Cluster power for $w_{2}=0.75$ and $w_{3}=0.25$.

**Figure 16 sensors-24-05851-f016:**
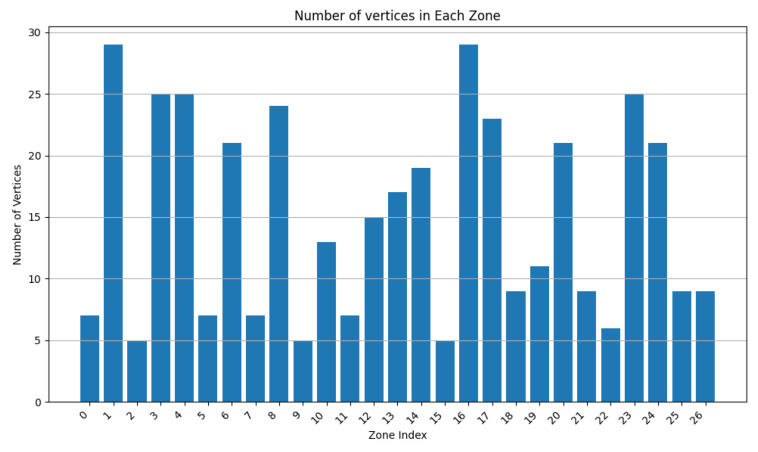
Cluster vertex number for $w_{2}=0.75$ and $w_{3}=0.25$.

**Figure 17 sensors-24-05851-f017:**
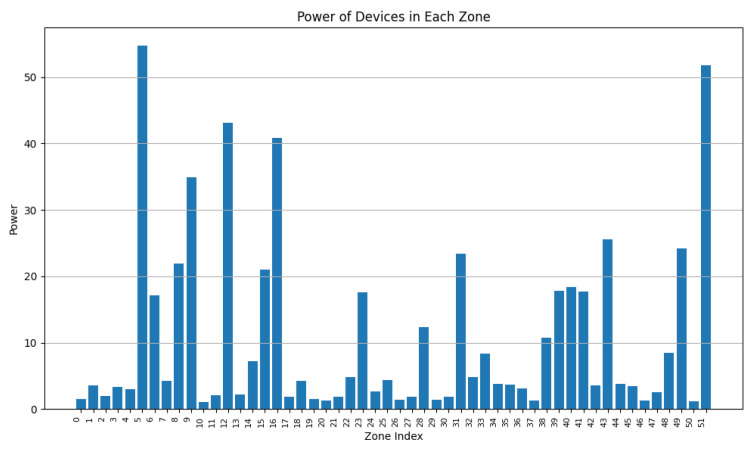
Cluster power for $w_{2}=0.5$ and $w_{3}=0.5$.

**Figure 18 sensors-24-05851-f018:**
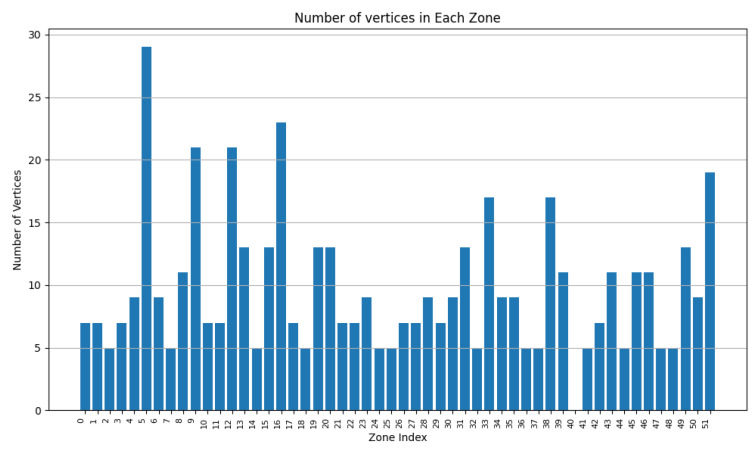
Cluster vertex number for $w_{2}=0.5$ and $w_{3}=0.5$.

**Figure 19 sensors-24-05851-f019:**
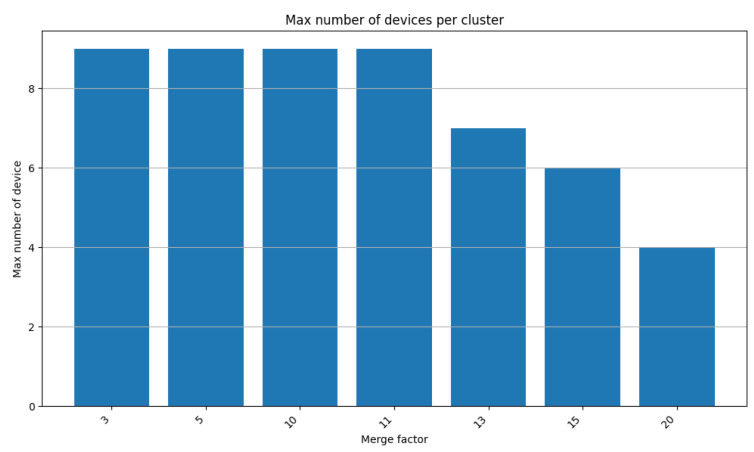
Merge factor impact.

**Figure 20 sensors-24-05851-f020:**
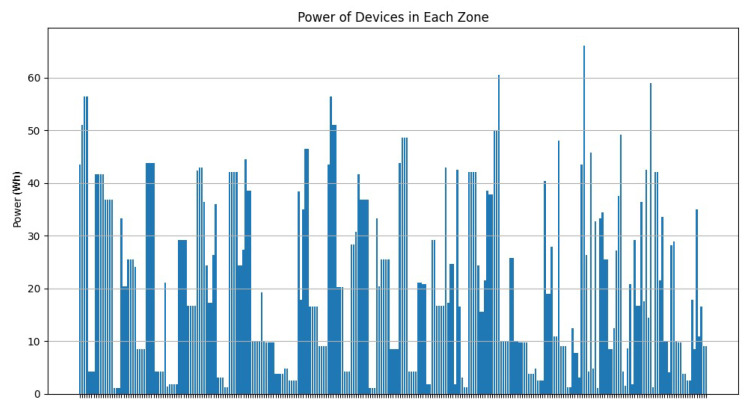
RDSC device power result for 1000 devices.

**Figure 21 sensors-24-05851-f021:**
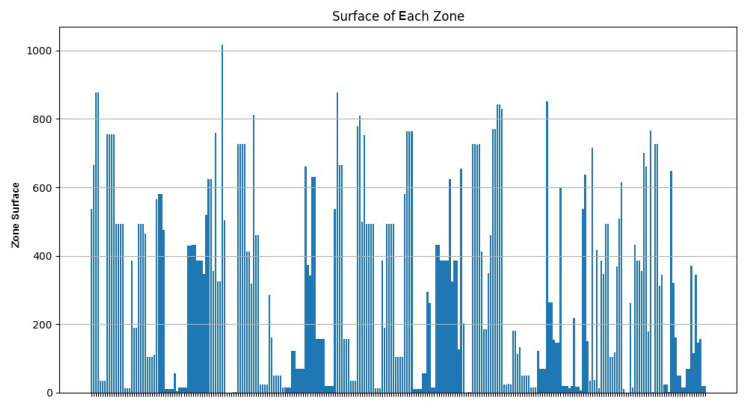
RDSC zone surface result for 1000 devices.

**Figure 22 sensors-24-05851-f022:**
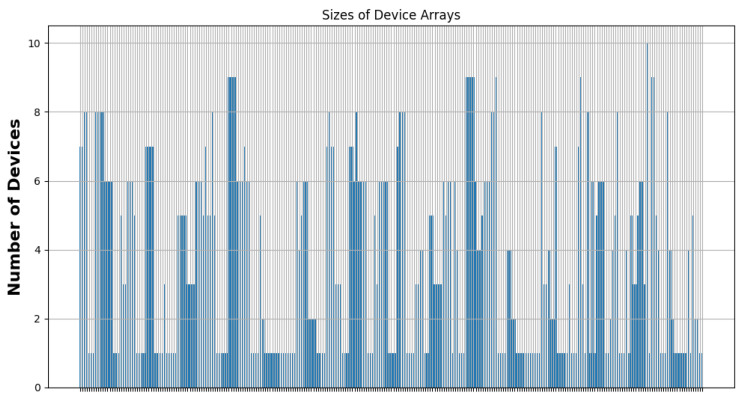
RDSC device numbers per cluster result for 1000 devices.

**Figure 23 sensors-24-05851-f023:**
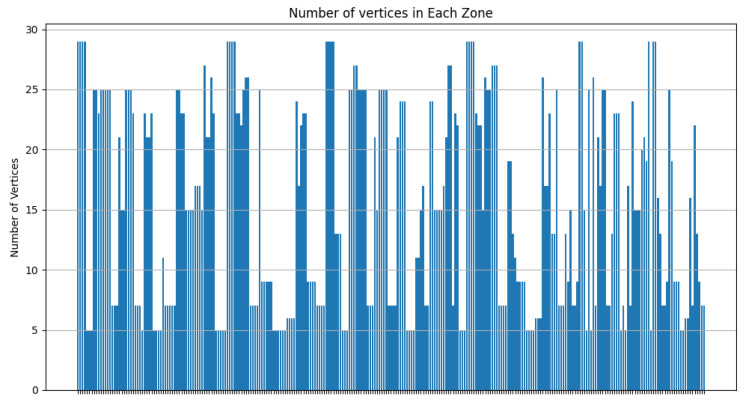
RDSC number of vertices’ result for 1000 devices.

**Figure 24 sensors-24-05851-f024:**
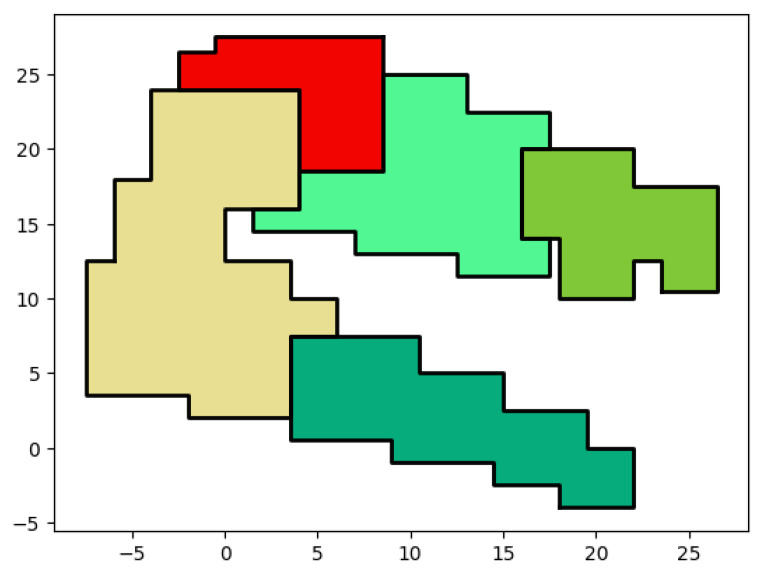
RDSC result for 20 devices.

**Figure 25 sensors-24-05851-f025:**
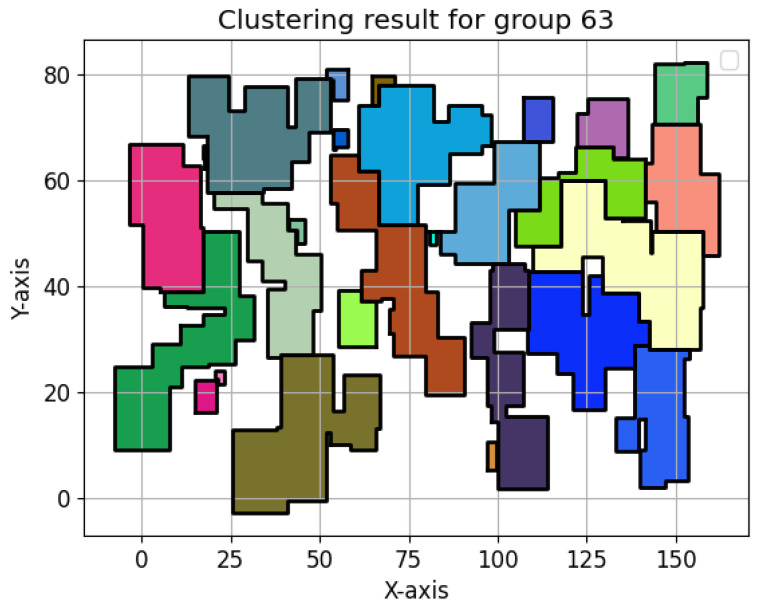
RDSC result for 100 devices.

**Figure 26 sensors-24-05851-f026:**
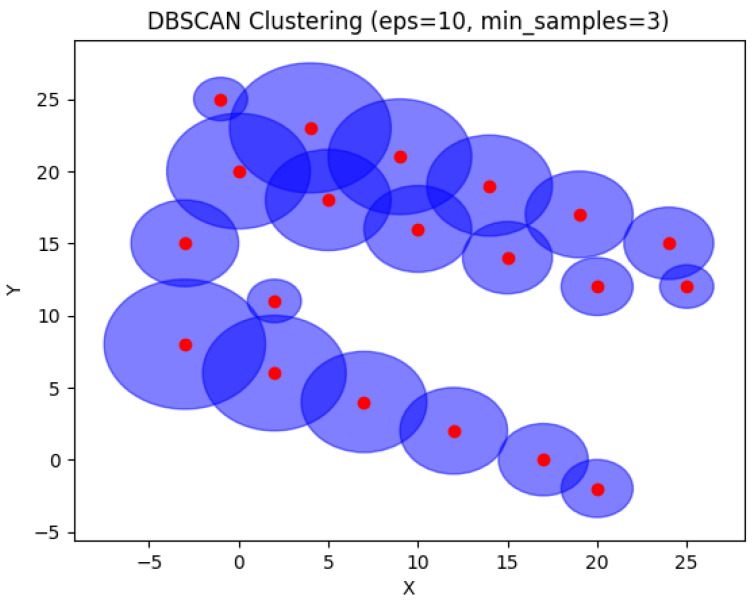
DBSCAN result for 20 devices.

**Figure 27 sensors-24-05851-f027:**
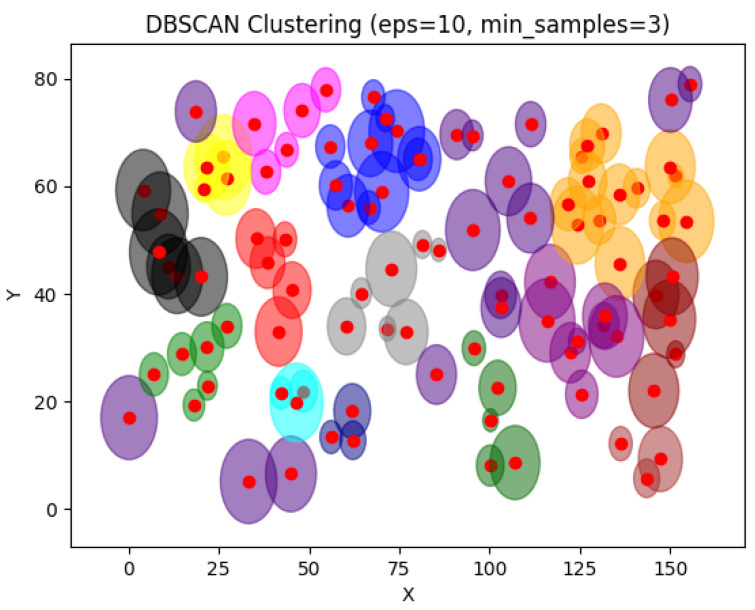
DBSCAN result for 100 devices.

**Figure 28 sensors-24-05851-f028:**
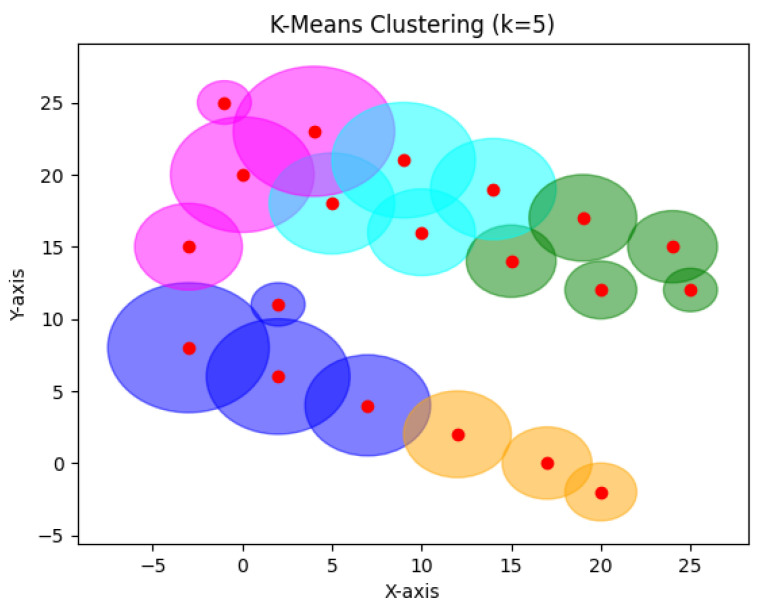
K-means result for 20 devices.

**Figure 29 sensors-24-05851-f029:**
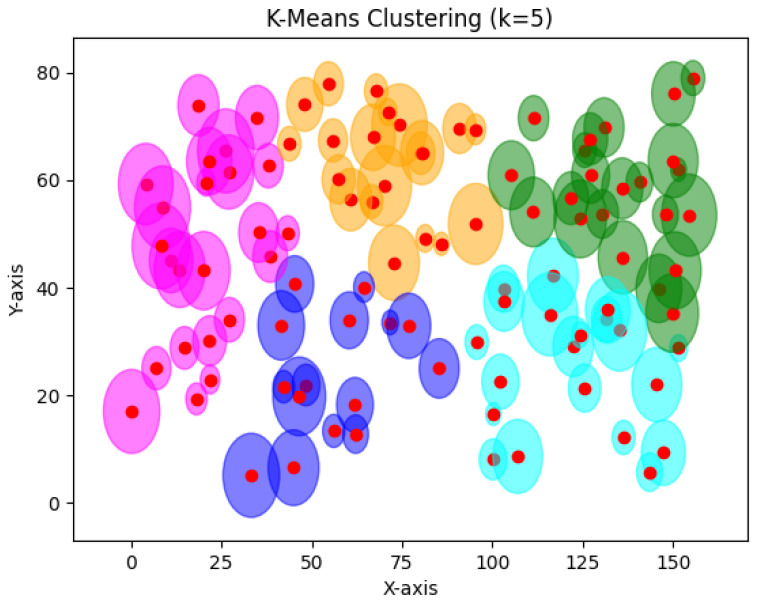
K-means result for 100 devices.

**Table 1 sensors-24-05851-t001:** Comparison table of clustering approaches for IoT resources (devices).

Contribution	Coverage Range	Energy/Power	Device Capacity
Essalhi et al. [28]	-	✓	✓
Ruha et al. [19]	-	-	✓
Saif et al. [21]	-	✓	-
Mukherjee et al. [22]	-	✓	-
Basavaraj et al. [24]	✓	✓	-
El-Sharkawi et al. [20]	✓	-	-
Lin et al. [23]	-	✓	-
Rehman et al. [26,27]	-	✓	✓
Our approach	✓	✓	✓

**Table 2 sensors-24-05851-t002:** Variation of the surface weight parameter.

	Case 1	Case 2	Case 3
$w_{1}$	0.9	0.75	0.5
$w_{2}$	0	0	0
$w_{3}$	0.1	0.25	0.5
mergeF	3.5	3.5	3.5
deletionR	0.3	0.3	0.3

**Table 3 sensors-24-05851-t003:** Variation of the power weight parameter.

	Case 1	Case 2	Case 3
$w_{1}$	0	0	0
$w_{2}$	0.9	0.75	0.5
$w_{3}$	0.1	0.25	0.5
mergeF	3.5	3.5	3.5
deletionR	0.3	0.3	0.3

**Table 4 sensors-24-05851-t004:** Variation of the mergeF parameter.

	Case 1	Case 2	Case 3	Case 4	Case 5	Case 6	Case 7
$w_{1}$	0.4	0.4	0.4	0.4	0.4	0.4	0.4
$w_{2}$	0.3	0.3	0.3	0.3	0.3	0.3	0.3
$w_{3}$	0.3	0.3	0.3	0.3	0.3	0.3	0.3
mergeF	3	5	10	11	13	15	20
deletionR	0.3	0.3	0.3	0.3	0.3	0.3	0.3

**Table 5 sensors-24-05851-t005:** List of algorithms’ parameters.

RDSC	DBSCAN	K-Means
Surface weight: 0.3	EPS: 10	K: 5
Power weight: 0.4	Min samples: 3	-
Vertices weight: 0.3		
Merge factor: 3.5		
Deletion rate: 0.3		

## Data Availability

Dataset available on request from the authors.

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
