# Peer review of "RDSC: Range-Based Device Spatial Clustering for IoT Networks"

_sensors, 2024, doi:10.3390/s24175851_

Round 1

Reviewer 1 Report

Comments and Suggestions for Authors

This paper proposes a device spatial clustering technique to address crucial challenges in IoT. The approach considers the coverage range of devices, improving data collection and aggregation processes. Additionally, it takes into account devices' storage capacities and power consumption, enhancing network lifetime and reducing storage costs for uncovered areas. Following are my comments.

1   The abstract needs enhancement to focus clearly on the problem and the proposed solution. Additionally, include the percentage of improvements achieved by the proposed work

2)   The authors have made several claims in the introduction section about edge computing, automated device management, etc., but no references are provided. Why is this the case?

3)   Include a paragraph in the introduction section that briefly describes the problems present in existing works.

4)  There are several state-of-the-art schemes available that target the issue of network lifetime enhancement in IoT. How is this work different and novel compared to others?

5) Provide a separate table for simulation parameters.

6) Highlight the physical and MAC layer parameters used in the simulation environment.

7) Compare the proposed work with existing schemes.

8) The most recent works should be cited, such as:

   1) "Improving resource-constrained IoT device lifetimes by mitigating redundant transmissions across heterogeneous wireless multimedia of things."

   2) M. A. U. Rehman, R. Ullah, B. -S. Kim, B. Nour, and S. Mastorakis, "CCIC-WSN: An Architecture for Single-Channel Cluster-Based Information-Centric Wireless Sensor Networks," in IEEE Internet of Things Journal.

9) The paper requires significant improvements in terms of English grammar and flow.

10) References are not appropriate and lack relevancy to the provided work.

Comments on the Quality of English Language

Extensive improvement is required

Author Response

Thanks a lot for the nice feedback we got. It allowed us to improve the paper.

Comment 1:   The abstract needs enhancement to focus clearly on the problem and the proposed solution. Additionally, include the percentage of improvements achieved by the proposed work

Response 1:  We revised the abstract by highlighting the problems and the solutions proposed in this study. Our comparison was made to compare standard clustering algorithm results with ours which have many differences (removed overlaps, storage capacity criteria). The power was a parameter to reduce and increase the power inside the cluster during the clustering process.

Comment 2:   The authors have made several claims in the introduction section about edge computing, automated device management, etc., but no references are provided. Why is this the case?

Response 2:  We thought it is a secondary issue. In the revised version, we added the following review that contains different application domains for IoT: Kashani, M.H.; Madanipour, M.; Nikravan, M.; Asghari, P.; Mahdipour, E. A systematic review of IoT in healthcare: Applications, 723 techniques, and trends. Journal of Network and Computer Applications 2021, 192, 103164.

Comment 3:  Include a paragraph in the introduction section that briefly describes the problems present in existing works.

Response 3: We added a  paragraph describing the challenges that are developed in the upcoming sections.

Comment 4:  There are several state-of-the-art schemes available that target the issue of network lifetime enhancement in IoT. How is this work different and novel compared to others?

Response 4: Our approach extends network lifetime by grouping the according devices reducing data redundancy by dividing the connected environment into non-overlapping zones. In addition, the creation of uncovered zones will facilitate networking tasks hence increasing network lifetime. For example, while indexing the devices in a connected environment, adding information about uncovered zones can extend network lifetime by reducing the number of hops needed to collect information about that zone (a device having information about the uncovered zones can directly return a response instead of forwarding the packet to all the covered zones deducing that it is an uncovered zone).

Comment 5:  Provide a separate table for simulation parameters.

Response 5: we added tables for each experiment where parameter variation is required.

Comment 6:  Highlight the physical and MAC layer parameters used in the simulation environment.

Response 6: We added the following sentence: "For data transmission, we consider that devices have the required parameters to transfer the messages over wifi channels (frequency band equal to 2.4 GHz, transmission power equal to 20 dBm). We also consider that the IoT devices use the CSMA/CA access method avoiding collisions."

Comment 7:  Compare the proposed work with existing schemes.

Response 7: We don't understand here what's requested. Inessence, a comparison was made to compare standard clustering algorithm results with ours which have many differences. In clustering algorithms, the clustering of the devices does not take into consideration their coverage range. For approaches that took into consideration the coverage range of the devices, they did not consider overlaps and the storage capacity of the devices. This was demonstrated when comparing our approach to K-means and DBSCAN where overlaps can be clearly seen and huge clusters were created.

Comment 8:  The most recent works should be cited, such as:    1) "Improving resource-constrained IoT device lifetimes by mitigating redundant transmissions across heterogeneous wireless multimedia of things." and   2) M. A. U. Rehman, R. Ullah, B. -S. Kim, B. Nour, and S. Mastorakis, "CCIC-WSN: An Architecture for Single-Channel Cluster-Based Information-Centric Wireless Sensor Networks," in IEEE Internet of Things Journal.

Response 8: Thank you for these interesting references. We explored them and added them to our paper in the related works and comparison table.

Comment 9:  The paper requires significant improvements in terms of English grammar and flow.

Response 9: We did a deep and detailed checkup and corrected several forgotten grammar mistakes.

Comment 10:  References are not appropriate and lack relevancy to the provided work.

Response 10: We revised by removing some references in the clustering background and we several new appropriate references.

Reviewer 2 Report

Comments and Suggestions for Authors

Dear authors, first of all, I would like to inform that I appreciate the attention you have given to the article. You have provided a manuscript describing a new clusterization process of Internet of Things sensing devices including their sensing area amongst power consumption and storage capability.

However, I still have found space for improvements in your manuscript, specifically regarding its structure and some key points.

Please consider the following points:

OVERALL:

- Assertive phrases unrelated to the core of your work/contributions should be supported by adding references, even when written outside the related works section (e.g., lines 16-17: ‘The number of IoT devices is increasing each year’). Please check all over the manuscript.

- There are no subtitles for subfigures. For instance: (a) and (b) in Figure 1. They should be included. Please check all over the manuscript.

- Describe figures, tables, equations and algorithms before presenting them. This was specifically critical when reading the results in section 8.1.1.

- Include the axis units when dealing with dimensional variables (area, power, etc.).

- How the results presented can support the choice of your method over an existing one? Please highlight it. A novelty should be justified by an advantage over the existing scenario.

SPECIFIC POINTS:

- The abstract needs to be rewritten: the presented abstract resembles an introduction to the paper, as it describes the context without specifying the steps taken to solve the problem and address the existing scientific gaps. An abstract should be concise and objective, providing a brief overview of your framework, including highlights of the quantitative results obtained.

- In section 2.2, the authors mention that their devices are "heterogeneous with different parameters". Still, further ahead in lines 110-111, they state that they are working with thermometers: "In our study, devices sense the same observation (i.e., temperature)". Can you please clarify this point? In what sense are you referring to the heterogeneity (sensor model, platform, communication protocol)?

- In Table 1, instead of putting reference brackets, the authors should write the respective methods' names or the author's names (why not both?).

- The authors provided Table 1 as a comparison to highlight the scientific gap they are filling with their new clustering method. As the term ‘coverage range’ was used initially, in an IoT network context this term led me to confusion: many clustering techniques already consider the link range in the clustering process, so how could this be a novelty? However, after Table 1, the authors clarify that their criterion for 'coverage range' is the ‘sensor range’. This clarification appears for the first time too late. Thus, to enhance clarity and text flow, I would recommend describing your method and the concepts involved before comparing its details to other works.

- Using a consistent term or definition (‘sensor range’ or ‘sensing range’ for 'coverage range') throughout the paper enhances its "readability".

- Can the authors please elaborate on how do they determine the sensing range (R) of a sensor? 

- What was the reduction in power consumption of your method? The authors provided the energy in each cluster, in Wh, but there is no information on an eventual better performance when comparing to other methods. (e.g. a residual power graph would be positive).

- I have failed to identify a Time Complexity discussion regarding your method. Can you please elaborate on why it was not made, or – if it is the case – why it was not relevant?

Comments on the Quality of English Language

Although I am not a native English speaker, I noticed a few areas where the writing could be improved. Therefore, I would recommend an additional round of English proofreading and editing.

Author Response

Thanks a lot for the detailed and constructive feedback.

OVERALL:

Comment 1: Assertive phrases unrelated to the core of your work/contributions should be supported by adding references, even when written outside the related works section (e.g., lines 16-17: ‘The number of IoT devices is increasing each year’). Please check all over the manuscript.

Response 1: We added as proposed several references. For instance, we added a reference that contains the different IoT domain of application in the introduction. We also added more references demonstrating the IoT device clustering. We also removed some references from the clustering background part.

Comment 2: There are no subtitles for subfigures. For instance: (a) and (b) in Figure 1. They should be included. Please check all over the manuscript.

Response 2: We added subtitles for figures 1 and 4

Comment 3: Describe figures, tables, equations and algorithms before presenting them. This was specifically critical when reading the results in section 8.1.1.

Response 3: Thanks. We explained the graph in subsections 8.1.1 and 8.1.2. In addition, we added a table showing the variation of the parameters in all sections to make the explanation clearer.

Comment 4: Include the axis units when dealing with dimensional variables (area, power, etc.).

Response 4: We include the units for the power graphs(Wh). For the surface, the unit is not important (it can be in cm, m or Km)

Comment 5: How the results presented can support the choice of your method over an existing one? Please highlight it. A novelty should be justified by an advantage over the existing scenario.

Response 5: We don't really understand what to do here. A comparison was made to compare standard clustering algorithm results with ours which have many differences. In clustering algorithms, the clustering of the devices does not take into consideration their coverage range. For approaches that took into consideration the coverage range of the devices, they did not consider overlaps and the storage capacity of the devices. This was demonstrated when comparing our approach to K-means and DBSCAN where overlaps can be clearly seen and huge clusters were created.

SPECIFIC POINTS:

Comment 1: The abstract needs to be rewritten: the presented abstract resembles an introduction to the paper, as it describes the context without specifying the steps taken to solve the problem and address the existing scientific gaps. An abstract should be concise and objective, providing a brief overview of your framework, including highlights of the quantitative results obtained.

Response 1: Thank you for your remark. We revised the abstract accordingly

Comment 2: In section 2.2, the authors mention that their devices are "heterogeneous with different parameters". Still, further ahead in lines 110-111, they state that they are working with thermometers: "In our study, devices sense the same observation (i.e., temperature)". Can you please clarify this point? In what sense are you referring to the heterogeneity (sensor model, platform, communication protocol)?

Response 2: Heterogeneous means they can sense many attributes but for the sake of simplification  we showed in the paper how this can be done with only one attribute. We revised this in the paper to provide more clarification.

Comment 3: In Table 1, instead of putting reference brackets, the authors should write the respective methods' names or the author's names (why not both?).

Response 3: Not all current approaches have been named. Instead, we added the names of their corresponding authors

Comment 4: The authors provided Table 1 as a comparison to highlight the scientific gap they are filling with their new clustering method. As the term ‘coverage range’ was used initially, in an IoT network context this term led me to confusion: many clustering techniques already consider the link range in the clustering process, so how could this be a novelty? However, after Table 1, the authors clarify that their criterion for 'coverage range' is the ‘sensor range’. This clarification appears for the first time too late. Thus, to enhance clarity and text flow, I would recommend describing your method and the concepts involved before comparing its details to other works.

Response 4: We revised accordingly. We added the following sentence “We note that we define the coverage area as the sensor's sensing area and not its communication range.” before defining the challenges to make the ideas clearer.

Comment 5: Using a consistent term or definition (‘sensor range’ or ‘sensing range’ for 'coverage range') throughout the paper enhances its "readability".

Response 5: We added this sentence to the paper before the challenges ‘ We will use the terms "sensor range" and "sensing range" interchangeably to refer to the coverage area.’

Comment 6: Can the authors please elaborate on how do they determine the sensing range (R) of a sensor?

Response 6: The sensing range of a sensor is determined by referring to the specifications provided by the sensor manufacturers. These specifications typically include the sensor's range, which is based on the sensor type and its design characteristics. By consulting the manufacturer's documentation, one can accurately identify the sensing range for each sensor.

Comment 7: What was the reduction in power consumption of your method? The authors provided the energy in each cluster, in Wh, but there is no information on an eventual better performance when comparing to other methods. (e.g. a residual power graph would be positive).

Response 7: Power consumption is affected by most of the attributes of our algorithm (power weight, merge factor…) that have a direct effect on packet forwarding inside and between clusters. On the other hand, we added an explanation that clarifies this idea for future readers. In our approach, the parameters are very important. The number of devices inside the cluster, their coverage range, and the weights have significant effects on the final result. For example, by choosing the correct parameters, we can reduce the power consumption of the devices by choosing the correct power weight, merge factor, and deletion rates. For other environments, by choosing the correct parameters, the coverage range can be greatly enhanced compared to the number of devices. In other cases, a better device distribution can be achieved by varying the merge factor. In other words, the parameters must be adapted to the user's needs.

Comment 8: I have failed to identify a Time Complexity discussion regarding your method. Can you please elaborate on why it was not made, or – if it is the case – why it was not relevant?

Response 8: Our algorithm has two external loops while j is set to 0 to recheck for bypassed intersections After zone merging. The complexity is demonstrated in section 8 at the beginning n^2 log n. We added some explanation concerning the complexity: "The complexity of the algorithm is n^2 log n  because of the two external loops each loop is counted as n, while is being set to 0 to recheck for bypassed intersections between zone (since after a merge the number of zones is reduced, we multiplied n^2 by log n)".

Reviewer 3 Report

Comments and Suggestions for Authors

1- In abstract section:

A) write the program used in this study.

b) Explain what does the author proposes clearly?

c) Add quantitative results that reflect paper contribution.

2-Add references to introduction section, it is not contain any reference and this is not acceptable.

3-many equations are written without references also many variables meaning not appeared.

4- Eq. 8 need to be checked, is it right?

5- Explain how does the author extract eqs.(9-10-11).

6- Explain how the author specifies the values of w1 and w3?, what is the boundaries of them ?

7-Table 2 is not completed, check this table.

8- In conclusion section add a percentage values for the improvements achieved by this study based on results obtained.

Author Response

Thanks a lot for the detailed feedback.

Comment 1: In abstract section: A) write the program used in this study, b) Explain what does the author proposes clearly?, c) Add quantitative results that reflect paper contribution.

Response 1: Thank you for your remark. We rewrote the abstract taking into consideration your points. For the quantitative results, we compared our approach with K-means and DBSCAN.

Comment 2: Add references to introduction section, it is not contain any reference and this is not acceptable.

Response 2: Thank you for your remark. We added the following reference containing the different applications of IoT: "Kashani, M.H.; Madanipour, M.; Nikravan, M.; Asghari, P.; Mahdipour, E. A systematic review of IoT in healthcare: Applications, 723 techniques, and trends. Journal of Network and Computer Applications 2021, 192, 103164".

Comment 3: many equations are written without references also many variables meaning not appeared.

Response 3: We added the following URI as a reference to demonstrate the min-max normalization formula https://www.codecademy.com/article/normalization. We also adapted the formula to this reference (removed + xmin). This will not affect our calculations since xmin is equal to 0.

For the equation of the zone use cases, we added that a and b are zones. We also added that x in the min-max normalization is a numerical value between xmin and xmax.

Comment 4: Eq. 8 need to be checked, is it right?

Response 4: We double-checked equation 8. We added an equal for the less and greater signs. This equation means that the sum of the weights must be equal to 1 and each weight can be between 0 and 1.

Comment 5: Explain how does the author extract eqs.(9-10-11).

Response 5: This is our explanation for the zone extraction. We also added it to the article.

In equation 9: After applying the zone dominance, we will obtain two zones. One unchanged zone with its objective value. Another zone that has been cut reducing its surface and increasing its vertices. To obtain a single value for these zones we added their objective function values. In equation 10: After merging(unifying) zones we can easily calculate their combined objective value. In equation 11: Having two zones a and b we subtract the loss of zone b (f(b)*deletion rate) from zone a. 

Comment 6: Explain how the author specifies the values of w1 and w3?, what is the boundaries of them?

Response 6: We added to our experiment’s discussion and conclusion that the values of the weights directly affects the algorithm’s result this is why they must be chosen wisely. This is out of scope of the current study but will be addressed in a future work.

Comment 7: Table 2 is not completed, check this table.

Response 7: The table contains the parameters that are used with other clustering algorithms to compare their results with ours. To reduce this confusion, we added a table for each experiment where many variations of values are required to make the ideas clearer.

Comment 8: In conclusion section add a percentage values for the improvements achieved by this study based on results obtained.

Response 8: Our comparison was made to compare standard clustering algorithm results with ours which have many differences. The power was a parameter to reduce and increase the power inside the cluster during the clustering process. On the other hand, we added in our conclusion that this comparison was made because it was not mentioned in the conclusion.

Reviewer 4 Report

Comments and Suggestions for Authors

The paper presents a significant contribution to the field of IoT device management through the presentation and application of the RDSC clustering method. The detailed methodology and rigorous testing enhance its credibility and relevance. With some adjustments based on the suggestions provided, this paper has the potential to make a strong impact in the emerging field of IoT network optimization.

  • It may be helpful to summarize the limitations of the discussed works more explicitly and indicate how the presented work directly contributes to solving these limitations.
  • The authors are invited to add a flowchart for decision-making to clarify the processes.
  • It would be helpful if the authors discuss the possible limitations of the method and the limitations in the experiments
  • The authors are invited to indicate future works following this research work. A way of optimizing the parameters and an automated way to find the appropriate configurations could be a good idea.
Comments on the Quality of English Language

Few typos to be corrected

Author Response

Thanks a lot for the different inputs. This allowed us to improve the paper.

Comment 1: It may be helpful to summarize the limitations of the discussed works more explicitly and indicate how the presented work directly contributes to solving these limitations.

Response 1:  Thank you for your remark. We added the following part to our discussion: "In these experiments, we demonstrated that our approach clustered the devices taking into account their storage capacity by reducing the number of vertices. It also partitioned their coverage range while removing intersection areas. Last but not least, the power is taken into consideration by choosing to minimize or maximize the power impact on the result.

Comment 2: The authors are invited to add a flowchart for decision-making to clarify the processes.

Response 2: We added a flowchart demonstrating the flow in section 6.2

Comment 3: It would be helpful if the authors discuss the possible limitations of the method and the limitations in the experiments
Response 3: Thank you for you remark. We added the following text in our discussion: "Following these experiments, the parameters must be chosen wisely which can be challenging in some cases.",
"In addition, connectivity must be taken into consideration to enhance the network performance."

Comment 4: The authors are invited to indicate future works following this research work. A way of optimizing the parameters and an automated way to find the appropriate configurations could be a good idea.

Response 4: Thank you for your note. We already mentioned in our conclusion that one of our future work is Parameter recommendation techniques to be implemented in future works to help users to find appropriate configurations depending on their current needs and environmental conditions.
Thus, we added an additional  future work in the conclusion:
"Moreover, as a future work, we will consider the connectivity to enhance the network performance."

Round 2

Reviewer 2 Report

Comments and Suggestions for Authors

Dear authors, thank you for providing a revised version of your manuscript. I have identified several improvements in overall quality, either from my points or my fellow referees.

However, I still have found space for improvement in two following points:

Overall, response 5:
I understand that you have compared your approach with standard clustering algorithms like K-means and DBSCAN, highlighting the differences between them and your method.

However, it is very important for readers to understand these advantages early in the article. I have noted that you included a sentence within the abstract where readers can grasp the main point of your paper. But I would still suggest including a specific section, or paragraph, as sooner as possible where you clearly explain how your approach outperforms existing methods, particularly in terms of coverage, overlap, and storage capacity of the devices (like you did in Conclusions, between lines 737-741). For example, you could emphasize that, unlike traditional methods, your approach considers coverage and avoids overlaps, resulting in more efficient and smaller clusters. This will help justify the novelty and relevance of your contribution from the beginning of the text.

Specific, Response 6: 
Regarding the coverage area and sensors, I understand that thermal cameras can detect temperatures from a distance. However, in the case of electronic temperature sensors (which are the sensors you are considering), the concept of a ‘coverage area’ is not typically provided by manufacturers. Temperature sensors (e.g., bandgap sensors) usually measure the temperature at a specific point or, at best, within a very small vicinity around the sensor, rather than covering a large area.

Could you please provide more details on how you determine the coverage area for temperature sensors in your algorithm? Additionally, please be mindful of the terminology used; the term ‘sensing range’ usually refers to the measurement scope of a sensor (e.g., -50°C to 150°C). I strongly recommend using ‘coverage area’ in all instances where applicable.

Thank you for your attention to this matter. I look forward to your clarifications.

Good work!

Author Response

Again, thanks for the constructive feedback and inputs. Hope you will be satisfied now by our following answers/changes:

Comment 1:  I understand that you have compared your approach with standard clustering algorithms like K-means and DBSCAN, highlighting the differences between them and your method. However, it is very important for readers to understand these advantages early in the article. I have noted that you included a sentence within the abstract where readers can grasp the main point of your paper. But I would still suggest including a specific section, or paragraph, as sooner as possible where you clearly explain how your approach outperforms existing methods, particularly in terms of coverage, overlap, and storage capacity of the devices (like you did in Conclusions, between lines 737-741). For example, you could emphasize that, unlike traditional methods, your approach considers coverage and avoids overlaps, resulting in more efficient and smaller clusters. This will help justify the novelty and relevance of your contribution from the beginning of the text.

Response 1: Thank you for your first comment. You're right; it's crucial to highlight this from the beginning. That's why we've included the corresponding paragraph in the introduction: "A comparison was made between RDSC and other clustering techniques (K-means and DBSCAN). The comparison demonstrated that these clustering algorithms resulted in overlapping big clusters without considering the device storage capacities and the residual energy. For this reason, we propose the RDSC approach, which clusters devices based on their coverage range, eliminates overlaps, and takes into account both device storage capacities and residual power while clustering, emphasizing the novelty of our approach".

Comment 2:  Regarding the coverage area and sensors, I understand that thermal cameras can detect temperatures from a distance. However, in the case of electronic temperature sensors (which are the sensors you are considering), the concept of a ‘coverage area’ is not typically provided by manufacturers. Temperature sensors (e.g., bandgap sensors) usually measure the temperature at a specific point or, at best, within a very small vicinity around the sensor, rather than covering a large area. Could you please provide more details on how you determine the coverage area for temperature sensors in your algorithm? Additionally, please be mindful of the terminology used; the term ‘sensing range’ usually refers to the measurement scope of a sensor (e.g., -50°C to 150°C). I strongly recommend using ‘coverage area’ in all instances where applicable.

Response 2: Thanks for this input. To avoid ambiguity, we revised the terminology by replacing all instances of "sensing range" and "sensor range" by "coverage range." We also noted our illustration with infrared sensors. In fact, those later are non-contact sensors that can measure temperature in distance in a specific area. So, we added the following paragraph in the chiberta forest setup section that concerns the determination of the coverage range: "Infrared sensors are non-contact sensors that can measure temperature within a specific coverage range. The coverage range can vary depending on the sensor type, the detector sensitivity, and the intensity of the emitted radiation. Typically, infrared sensors can measure distances ranging from a few centimeters to several meters".